# Physical constraints and biological regulations underlie universal osmoresponses

Yiyang Ye[1], Qirun Wang[1], Jie Lin[1,2]*

[1]Center for Quantitative Biology, Academy for Advanced Interdisciplinary Sudies, Peking University, Beijing, China; [2]Peking-Tsinghua Center for Life Sciences, Academy for Advanced Interdisciplinary Sudies, Peking University, Beijing, China

## eLife Assessment

This manuscript develops a theoretical model of osmotic pressure adaptation in microbes by osmolyte production and wall synthesis. The prediction of a rapid increase in growth rate on osmotic shock is experimentally validated using fission yeast. By using phenomenological rules rather than detailed molecular mechanisms, the model can potentially apply to a wide range of microbes, providing **important** insights that would be of interest to the wider community studying the regulation of cell size and mechanics. The level of coarse-graining and the assumptions and limitations of the model have been well described, providing a **convincing** foundation for making predictions. However, further experimental work on the validity of the core assumptions across a range of microbial organisms is needed to assess the universality of the model.

*For correspondence:
linjie@pku.edu.cn

Competing interest: The authors declare that no competing interests exist.

**Abstract** Microorganisms constantly transition between environments with dramatically different external osmolarities. However, theories of microbial osmoresponse integrating physical constraints and biological regulations are lacking. Here, we propose such a theory, utilizing the separation of timescales for passive responses and active regulations. We demonstrate that regulations of osmolyte production and cell-wall synthesis assist cells in coping with intracellular crowding effects and adapting to a broad range of external osmolarity. Furthermore, we predict a threshold value above which cells cannot grow, ubiquitous across bacteria and yeast. Intriguingly, the theory predicts a dramatic speedup of cell growth after an abrupt decrease in external osmolarity due to cell-wall synthesis regulation. Our theory rationalizes the unusually fast growth observed in fission yeast after an oscillatory osmotic perturbation, and the predicted growth rate peaks match quantitatively with experimental measurements. Our study reveals the physical basis of osmoresponse, yielding far-reaching implications for microbial physiology.

## Introduction

Microbes constantly transition between environments with dramatically different osmolarities, a hallmark of microbial life (*Csonka, 1989*; *Muzzey et al., 2009*; *Wood, 2015*; *Bremer and Krämer, 2019*). One of the most essential features of walled microbial cells is the turgor pressure – the elastic stress stretching the cell wall due to osmotic imbalance. Upon a hypoosmotic shock (i.e. a sudden decrease of the external osmolarity), the turgor pressure increases immediately due to the sudden water influx. To relax the turgor pressure, the cell upregulates the cell-wall synthesis rate, adds more materials to the peptidoglycan network, and eventually adapts to the lower external osmolarity (*Typas et al., 2010*). Upon a hyperosmotic shock (i.e. a sudden increase of the external osmolarity),

the cell volume of a microbial cell shrinks within milliseconds due to water efflux, leading to a decreased turgor pressure (*Rojas and Huang, 2018*; *Cadart et al., 2019*). To increase the internal osmotic pressure, microorganisms increase their intracellular solute pool by amassing osmolyte molecules (i.e. osmoregulation), e.g., through de novo synthesis (*Kempf and Bremer, 1998*). The cell volume then restores progressively over time, and eventually, the cell adapts to the higher osmolarity. Intracellular crowding may act as a cell volume sensor to trigger osmoregulation (*Burg, 2000*; *van den Berg et al., 2017*; *Model et al., 2021*). Meanwhile, intracellular crowding due to volume reduction inevitably affects the cellular physiology globally, e.g., slowing down protein diffusion (*Dix and Verkman, 2008*; *Dill et al., 2011*; *Miermont et al., 2013*; *Mika et al., 2014*; *Munder et al., 2016*; *Joyner et al., 2016*; *Molines et al., 2022*) and reducing the elongation speed of translating ribosomes (*Dai et al., 2018*; *Chen et al., 2023*). Despite extensive knowledge regarding the molecular details of osmotic response pathways (*Bremer and Krämer, 2019*), how intracellular crowding interferes with gene expression regulation and affects osmotic adaptation remains an open question.

Interestingly, many features of microbial osmoresponses appear general across different organisms, suggesting a universal underlying mechanism. For example, it is widely observed that microbial cells can adapt to a broad range of external osmolarity, with the external osmotic pressure varying over an order of magnitude (*Cayley et al., 1991*; *Dai et al., 2018*; *Rojas et al., 2014*; *Rojas et al., 2017*). Furthermore, the growth rate in the steady state decreases as the external osmolarity increases, and a complete arrest of cell growth occurs above a critical osmolarity (*Scott, 1953*; *Christian and Scott, 1953*; *Christian, 1955*; *Rojas et al., 2014*; *Rojas et al., 2017*; *Dai et al., 2018*). Moreover, upon an osmotic shock, the growth rate usually does not approach the new steady-state value monotonically, e.g., an overshoot of growth rate often occurs upon a hypoosmotic shock (*Rojas et al., 2017*), and a damped oscillation of growth rate can happen after a hyperosmotic shock (*Rojas et al., 2014*). In recent experiments of *Schizosaccharomyces pombe* by *Knapp et al., 2019*, an oscillatory osmotic shock was applied to cells, during which cell volume growth was dramatically slowed down while biomass was still actively produced. Surprisingly, a supergrowth phase happened after removing the oscillatory osmotic shock, during which cells grew much faster than the steady state before the shocks.

In this work, we unify all these phenomena by a theory capturing the essential elements of osmo-responses: physical constraints (e.g. the crowding effects and osmotic imbalance) and biological regulation, including osmoregulation (i.e. regulation of the osmolyte-producing protein) and cell-wall synthesis regulation. Our model assumes the following phenomenological rules: (1) the change in free water volume within the cell is driven by osmotic imbalance (*Cadart et al., 2019*; *Rollin et al., 2023*), while the remaining volume changes in proportion to protein production; (2) osmoregulation influences the production of osmolyte-producing protein, governed by intracellular protein density; (3) cell-wall synthesis is regulated through a feedback mechanism, wherein turgor pressure modulates the efficiency of cell-wall synthesis, enabling the cell to maintain a relatively stable turgor pressure; and (4) intracellular crowding slows down biochemical reactions as the protein density increases, with reactions ceasing entirely when the protein density reaches a critical threshold. Upon a hyperosmotic shock, cell volume reduction due to water efflux increases the protein density, inducing the upregulation of osmolyte-producing protein but slowing down the translation speed due to crowding. Upon a hypoosmotic shock, the dramatic water influx stretches the cell wall, and the increased turgor pressure induces cell-wall synthesis (*Typas et al., 2010*; *Jiang and Sun, 2010*; *Amir and Nelson, 2012*).

We remark that our model is coarse-grained, without including detailed molecular mechanisms, and is therefore applicable across diverse microbial species. Notably, the predicted steady-state growth rate as a function of internal osmotic pressure from our model aligns well with experimental data from diverse organisms. This alignment allows us to quantify the sensitivities of translation speed and regulation of osmolyte-producing protein in response to intracellular density. Additionally, we demonstrate that osmoregulation and cell-wall synthesis regulation enable cells to adapt to a wide range of external osmolarities and prevent plasmolysis. Our model also predicts a non-monotonic time dependence of growth rate and protein density as they approach steady-state values following a constant osmotic shock, in concert with experimental observations (*Rojas et al., 2014*; *Rojas et al., 2017*). Moreover, we show that a supergrowth phase can arise following a sudden decrease in external osmolarity, driven by cell-wall synthesis regulation, either through the direct application of a hypoosmotic shock or the withdrawal of an oscillatory stimulus. Remarkably, the predicted amplitudes of

supergrowth (i.e. growth rate peaks) quantitatively agree with multiple independent experimental measurements (*Knapp et al., 2019*).

In the following Results section, we begin by outlining the primary assumptions and equations of our model in the subsection Model description, which includes four parts, each addressing one of the four phenomenological rules. Additional details can be found in Materials and methods. We then proceed to the subsection Steady states in constant environments, where we employ our theoretical framework to analyze steady-state growth and examine how the growth rate varies with external osmolarity. In the subsection Transient dynamics after a constant osmotic shock, we investigate the time-dependent osmoresponse after a constant hyperosmotic and hypoosmotic shock. Finally, in Comparison with experiments: supergrowth phenomena after osmotic oscillation, we address the supergrowth phenomena observed in *S. pombe*, utilizing our model to elucidate these experimental observations.

## Results

### Model description

#### Cell growth

In the limit of an extreme hyperosmotic shock, the remaining cytoplasmic volume is comparable to the volume of expelled water (*Cayley et al., 1991*; *Scott Cayley et al., 2000*; *Miermont et al., 2013*). Thus, the total cytoplasmic volume must be divided into a free volume and a bound volume (*Whatmore and Reed, 1990*; *Cayley and Record, 2003*; *Lemière and Chang, 2023*; *Zhou et al., 2009*; *Rollin et al., 2023*):

$$V = V_f + V_b. \tag{1}$$

The free volume comes from the free water that is osmotically active, and the bound volume includes the bound water $V_{bw}$ (i.e. water of macromolecular hydration) and the volume of dry mass $V_{bd}$ (*Figure 1A*). Because the fraction of protein mass in the total dry mass is typically constant and the volume of bound water is proportional to the dry mass (*Cayley et al., 1991*), the bound volume is proportional to the total protein mass $m_p$ through $V_b = \alpha m_p$. Here, $\alpha$ is a constant, and its values for some model organisms are included in *Table 1*, and its detailed calculations from experimental data are in Section B of Appendix 1.

The free volume changes due to osmotic imbalance, and the growth rate of the free volume follows

$$\mu_f \equiv \frac{\dot{V}_f}{V_f} = k_w(\Pi_{in} - \Pi_{out} - \sigma), \tag{2}$$

where $\Pi_{in}$, $\Pi_{out}$ are the internal (i.e. cytoplasmic) and external osmotic pressures, respectively (*Cadart et al., 2019*). $\Pi_{in}$ is proportional to the concentration of osmolyte molecules in the free volume: $\Pi_{in} = k_B T N_a / V_f$, where $N_a$ is the number of osmolyte molecules, $k_B$ is the Boltzmann constant, and $T$ is the temperature. For simplicity, we assume that the production speed of osmolyte molecules is proportional to the mass of osmolyte-producing protein (Materials and methods). Here, we have replaced the difference of the hydrostatic pressures across the cell membrane with the turgor pressure $\sigma$, assuming that mechanical equilibrium is always satisfied. $k_w$ is the filtration coefficient quantifying the water permeability of the cell membrane (*Solenov et al., 2017*).

The species of osmolytes involved in osmoregulation are diverse across different microorganisms and conditions; nevertheless, they are primarily small organic molecules (*Kempf and Bremer, 1998*; *Empadinhas and da Costa, 2008*). In this work, we simplify the problem by considering a single species of osmolyte that dominates the internal osmotic pressure, e.g., glycerol in *Saccharomyces cerevisiae* (*Reed et al., 1987*; *Hohmann et al., 2007*; *Blomberg, 2022*) and glycine betaine in *Escherichia coli* (*Wood, 2015*), with the production speed proportional to the mass of the osmolyte-producing protein (*Figure 1A* and Materials and methods).

To model gene expression regulation, we introduce $\chi_a$ and $\chi_r$ as the fractions of ribosomes translating the osmolyte-producing protein and ribosomal proteins (*Figure 1B* and Materials and methods). In steady states, $\chi_a$ and $\chi_r$ are equal to the mass fractions of osmolyte-producing protein and ribosomal proteins in the total proteome, $\phi_a = m_{p,a}/m_p$ and $\phi_r = m_{p,r}/m_p$, respectively (*Scott et al.,*

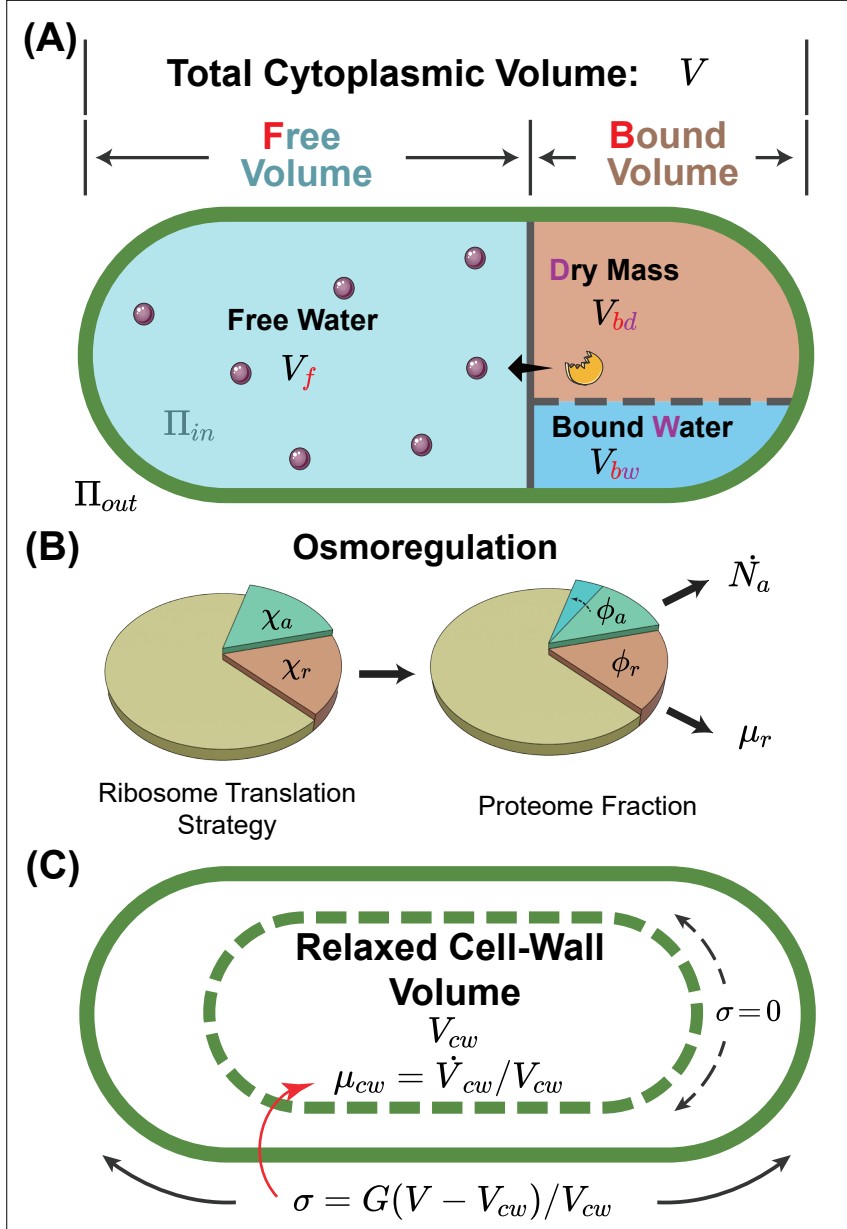

**Figure 1.** A schematic of the osmoresponse model. (**A**) The total cytoplasmic volume includes the free and bound volumes. The free volume sets the internal osmotic pressure $\Pi_{in} = k_B T N_a / V_f$, where $V_f$ is the free volume and $N_a$ is the number of osmolyte molecules. The bound volume $V_b$ comprises the dry mass $V_{bd}$ and bound water $V_{bw}$, i.e., $V_b = V_{bd} + V_{bw}$, all proportional to the total protein mass. (**B**) We model osmoregulation through the change of ribosome translation strategy. When the protein density increases, the fraction of ribosomes translating the osmolyte-producing protein $\chi_a$ is upregulated, leading to the subsequent increase in the mass fraction of the osmolyte-producing protein $\phi_a$. Here, $\mu_r$ denotes the dry-mass growth rate. (**C**) The cell-wall synthesis process is controlled by the turgor pressure $\sigma$, which is proportional to the cell-wall strain $\epsilon = (V - V_{cw})/V_{cw}$. Here, $V$ is the cytoplasmic volume, and $V_{cw}$ is the relaxed cell-wall volume.

*2010*; *Wang and Lin, 2022*). In this work, we assume that the dry-mass growth rate is proportional to the fraction of ribosomal proteins within the total proteome for simplicity, $\mu_r = k_r m_{p,r}/m_p = k_r \phi_r$. This assumption leverages the fact that ribosomes are responsible for producing all proteins. The proportionality coefficient $k_r$ encapsulates the efficiency of ribosomal activity, being proportional to the elongation speed of the ribosome. We remark that $k_r$ is influenced by the crowding effect, which

**Table 1.** Model parameters for different species in their corresponding reference growth media.

| E. coli | Value | Reference | | |
|---|---|---|---|---|
| $\sigma_c$ | 1 [atm] | **Rojas and Huang, 2018** | | |
| $\alpha$ | 1.68 [ml/g] | Deduce from **Scott Cayley et al., 2000** | | |
| | MBM (**Cayley et al., 1991**) | MOPS+ fructose (**Dai et al., 2018**) | MOPS+ glucose (**Dai et al., 2018**) | LB (**Rojas et al., 2014**) |
| $k_r^{\max}\chi_r$ | 0.743 [1/hr] | 0.776 [1/hr] | 1.14 [1/hr] | 2.05 [1/hr] |
| $\Pi_{in,c}$ | 1.54 [Osm] | 1.49 [Osm] | 1.61 [Osm] | 2.18 [Osm] |
| $H_r/(H_a + 1)$ | 1.68 | 1.30 | 1.18 | 2.72 |

| Bacillus subtilis (LB) | Value | Reference |
|---|---|---|
| $\sigma_c$ | 19 [atm] | **Whatmore and Reed, 1990** |
| | 2.52 [1/hr] | |
| $H_r/(H_a + 1)$ | 2.18 | |
| $\Pi_{in,c}$ | 3.09 [Osm] | Fit to **Rojas et al., 2017** |

| S. pombe (YE5S) | Value | Reference |
|---|---|---|
| $\Pi_{out}$ | 0.2 [Osm] | **Atilgan et al., 2015** |
| $\sigma_c$ | 10 [atm] | **Lemière and Chang, 2023** |
| $\hat{\rho}_d$ | 0.282 [g/ml] | **Odermatt et al., 2021** |
| $\rho_p$ | 0.104 [g/ml] | See Section B of Appendix 1 |
| $\mu$ | 0.35 [1/hr] | **Knapp et al., 2019** |
| $f$ | 0.788 | Fit to **Molines et al., 2022** (Section B of Appendix 1) |
| $\epsilon$ | 0.584 | **Atilgan et al., 2015** |
| $\alpha$ | 2.60 [ml/g] | See Section B of Appendix 1 |
| $G$ | 17.1 [atm] | $G = \sigma/\epsilon$ |
| $\Pi_{out,c}$ | 3.5 [Osm] | |
| $k_w$ | 100 [1/(min atm)] | |
| $\rho_c$ | 0.267 [g/ml] | Deduce from **Molines et al., 2022** (Section B of Appendix 1) |
| $H_r$ | 3.03 | Copied from S. cerevisiae in YPD |
| $H_a$ | 0.974 | Set according to $\Pi_{in}/\Pi_{in,c} = \left(\rho_p/\rho_c\right)^{H_a}$ |
| $k_r^{\max}\chi_r$ | 0.371 [1/hr] | Set according to $\mu_r = k_r^{\max}\chi_r\left(1 - \left(\rho_p/\rho_c\right)^{H_r}\right)$ |
| $k_B T k_a^{\max}\chi_a^{\max}$ | 2.25 [(atm ml)/(g min)] | Set according to $k_B T k_a^{\max}\chi_a^{\max}\eta_a\rho_p = k_r^{\max}\chi_r\Pi_{in}$ |
| $\tau_{cw}^-$ | 0.1 [min] | |
| $\tau_{cw}^+$ | 12.5 [min] | |
| $H_{cw}$ | 1.7 | Fit to **Knapp et al., 2019** |

| S. cerevisiae (YPD) | Value | Reference |
|---|---|---|
| $\Pi_{out}$ | 0.26 [Osm] | |

*Table 1 continued on next page*

*Table 1 continued*

| *S. cerevisiae* (YPD) | Value | Reference |
|---|---|---|
| $\sigma_c$ | 3.1 [atm] | *Lemière and Chang, 2023* |
| $\hat{\rho}_d$ | 0.295 [g/ml] | *Feijó Delgado et al., 2013* |
| $\mu$ | 0.448 [1/hr] | Our experiment |
| $f$ | 0.6 | *Miermont et al., 2013* |
| $\rho_p$ | 0.155 [g/ml] | See Section B of Appendix 1 |
| $\rho_c$ | 0.994 [g/ml] | See Section B of Appendix 1 |
| $\alpha$ | 4.29 [ml/g] | See Section B of Appendix 1 |
| $k_r^{\max} \chi_r$ | 0.450 [1/hr] | |
| $\Pi_{in,c}$ | 3.52 [Osm] | |
| $H_r/(H_a + 1)$ | 2.54 | Fit to our data |
| $H_r$ | 3.03 | Set according to $1 - (\rho_p/\rho_c)^{H_r} = \mu/(k_r^{\max}\chi_r)$ |

we address later. The growth rate of the cytoplasmic volume, $\mu = \dot{V}/V$, is a weighted average of the free-volume growth rate $\mu_f$ and the dry-mass growth rate $\mu_r$:

$$\mu = f\mu_f + (1 - f)\mu_r. \tag{3}$$

Here, $f$ is the free volume fraction in the total cytoplasmic volume: $f = V_f/V$. In this work, we refer to the growth rate as the growth rate of cytoplasmic volume $\mu$ unless otherwise mentioned.

## Osmoregulation

*Dai et al., 2018*, found that the reduction of growth rate as the external osmolarity increases is dominated by the reduction of the translation speed $k_r$ instead of the ribosomal fraction $\phi_r$. Therefore, we assume that the fraction of ribosomes translating themselves $\chi_r$ is constant for simplicity. To model osmoregulation, we introduce a coupling between the fraction of ribosomes translating the osmolyte-producing protein $\chi_a$ and the degree of intracellular crowding. We quantify the crowding effects by the protein density, defined as $\rho_p = m_p/V_f$, which serves as a good proxy for the dry-mass density measured in the experiments (*Feijó Delgado et al., 2013*; *Odermatt et al., 2021*) (see *Table 1* and the detailed discussion on the relations between the two densities in Section A of Appendix 1) and propose the following relation:

$$\chi_a = \chi_a^{\max} \left( \frac{\rho_p}{\rho_c} \right)^{H_a}. \tag{4}$$

Here, the parameter $H_a$ quantifies the sensitivity of osmoregulation to intracellular crowding. $\rho_c$ is the critical protein density above which intracellular processes are frozen, which we introduce later in *Equation 8*. Therefore, $\chi_a^{\max}$ represents the largest possible $\phi_a$ since all intracellular dynamics is frozen when $\rho_p > \rho_c$. We remark that our model can be directly generalized to cases where osmolyte molecules are extracted from the environment. One only needs to change the interpretation of the parameter $k_a$ in *Equation 17* from the synthesis rate to the uptake rate, and all the results are the same.

## Cell-wall synthesis regulation

In this work, the cell wall is regarded as a linear elastic material, where the turgor pressure is proportional to the elastic strain of the cell wall by a constant modulus $G$ such that

$$\sigma = G\epsilon = G \left( \frac{V}{V_{cw}} - 1 \right). \tag{5}$$

Here, $V_{cw}$ is the relaxed cell-wall volume (*Figure 1C*). When plasmolysis happens, the cell membrane detaches from the cell wall ($V < V_{cw}$), and the turgor pressure is zero. We introduce the growth rate of the relaxed cell-wall volume as $\mu_{cw} = \dot{V}_{cw}/V_{cw}$. Given that in the steady states of cell growth, $\mu_r = \mu_{cw}$, we write $\mu_{cw}$ in the following form without losing generality,

$$\mu_{cw} = \mu_r \eta_{cw}. \tag{6}$$

Here, $\eta_{cw}$ is a coarse-grained parameter modeling the active regulation of cell-wall synthesis, which we refer to as the cell-wall synthesis efficiency in the following.

Experiments suggested that turgor pressure induces cell-wall synthesis, e.g., through mechanosensors on cell membrane in *S. pombe* (*Dupres et al., 2009*; *Neeli-Venkata et al., 2021*), by increasing the pore size of the peptidoglycan network (*Typas et al., 2010*) and by accelerating the moving velocity of the cell-wall synthesis machinery in *E. coli* (*Amir and Nelson, 2012*). Guided by these ideas, we model the effects of turgor pressure on the time dependence of the cell-wall synthesis efficiency as

$$\dot{\eta}_{cw} = \frac{1}{\tau_{cw}^{\pm}} \left[ \left( \frac{\sigma}{\sigma_c} \right)^{H_{cw}} - \eta_{cw} \right]. \tag{7}$$

Here, $\sigma_c$ is a characteristic scale of turgor pressure depending on species. $\tau_{cw}^{+}$ ($\tau_{cw}^{-}$) is the relaxation timescale when the current $\eta_{cw}$ is below (above) its target value $\eta_{cw}^{st} = (\sigma/\sigma_c)^{H_{cw}}$. The former (latter) happens immediately after the cell is subject to a hypoosmotic (hyperosmotic) shock. In the extreme case of plasmolysis, the insertion of newly synthesized cell-wall materials is interrupted immediately due to the separation of the cell membrane and cell wall. Meanwhile, the upregulation of cell-wall synthesis rate presumably takes a longer time. For example, in fungi, where polarized growth is generally adopted, the upregulation of the cell-wall synthesis rate involves reorienting the polarisome complex to the growing tip, directing actin polarization, and delivering cell-wall synthesis machinery (*Kono et al., 2012*; *Haupt et al., 2018*). Therefore, we set $\tau_{cw}^{+} \gg \tau_{cw}^{-}$ in this work (see details of parameter values in *Table 1*).

## Intracellular crowding

Multiple experiments suggested the cytoplasm of bacteria, yeast, and mammalian cells resemble crowded colloidal suspensions in which the mobilities of biomolecules are significantly reduced compared with dilute solutions (*Miermont et al., 2013*; *Parry et al., 2014*; *Mika et al., 2014*; *Nishizawa et al., 2017*; *Ebata et al., 2023*), a signature of glass transition (*Hunter and Weeks, 2012*). Intracellular crowding affects biochemical processes globally, e.g., slowing down translation and intracellular signaling by suppressing protein diffusion (*Miermont et al., 2013*; *Parry et al., 2014*; *Mika et al., 2014*; *Dai et al., 2018*; *Molines et al., 2022*). Therefore, we assume that the speed of osmolyte production, translational elongation, and cell-wall synthesis are all slowed down by the same crowding factor:

$$\eta_r = 1 - \left( \frac{\rho_p}{\rho_c} \right)^{H_r}. \tag{8}$$

Here, $\rho_c$ is the critical protein density, and $H_r$ is a parameter to quantify the sensitivity of biochemical reactions to the intracellular density. For example, the translational elongation speed is suppressed by intracellular crowding through $k_r = k_r^{max} \eta_r$. Therefore, the dry-mass growth rate becomes $\mu_r = \mu_r^{max} \eta_r$, where we introduce $\mu_r^{max} = k_r^{max} \phi_r$.

The details of our model are summarized in Materials and methods, with five independent variables: the protein density $\rho_p$, the mass fraction of osmolyte-producing protein $\phi_a$, the internal osmotic pressure $\Pi_{in}$, the cell-wall strain $\epsilon$, and the cell-wall synthesis efficiency $\eta_{cw}$. For convenience, *Appendix 1—table 3* provides a comprehensive list of all symbols used in the main text along with their meanings.

## Steady states in constant environments

When cell growth reaches a steady state, the proportions of all components, including free water volume, cell mass, and cell-wall volume, must be constant relative to the total cell volume to ensure

homeostasis. Therefore, all growth rates in steady states of cell growth must be the same: $\mu_f = \mu_r = \mu_{cw}$. The consequence of cell-wall synthesis regulation can be seen directly from $\mu_{cw} = \mu_r$: the turgor pressure at steady states is constant, $\sigma = \sigma_c$. Experimentally, the cell-wall strain was measured by applying an acute hyperosmotic shock to induce plasmolysis, and it is approximately constant as the external osmolarity increases (*Misra et al., 2013*; *Rojas et al., 2014*), suggesting a constant turgor pressure independent of external osmolarity, in concert with our model assumptions. The internal osmotic pressure at steady states is related to the external osmotic pressure through *Equation 2*,

$$\Pi_{in} = \Pi_{out} + \sigma. \tag{9}$$

Here, we have neglected the term $\mu_f/k_w$. *Boer et al., 2011*, show that an abrupt water flux occurs within hundreds of milliseconds after an osmotic shock, from which we can estimate the water permeability as $k_w \sim 100 \ \text{min}^{-1}\text{atm}^{-1}$ considering an osmotic shock with an amplitude $\Delta\Pi_{out} = 1$ atm. Because the typical doubling times of microorganisms are from about 20 min to several hours, we estimate $\mu_f/k_w \sim 10 - 100$ Pa (*Ye and Verkman, 1989*; *Boer et al., 2011*), negligible compared with the typical cytoplasmic osmotic pressures, which can be several atmospheric pressures.

In steady states, the internal osmotic pressure is independent of time. Combining *Equation 4* and the dynamics of the internal osmotic pressure, *Equation 18c*, we find the relationships between the protein density, the internal osmotic pressure, and the growth rate in the steady states:

$$\frac{\Pi_{in}}{\rho_p^{H_a+1}} = \text{const}, \tag{10a}$$

$$\frac{\mu_r}{\mu_r^{\text{max}}} = 1 - \left(\frac{\Pi_{in}}{\Pi_{in,c}}\right)^{\frac{H_r}{H_a+1}}. \tag{10b}$$

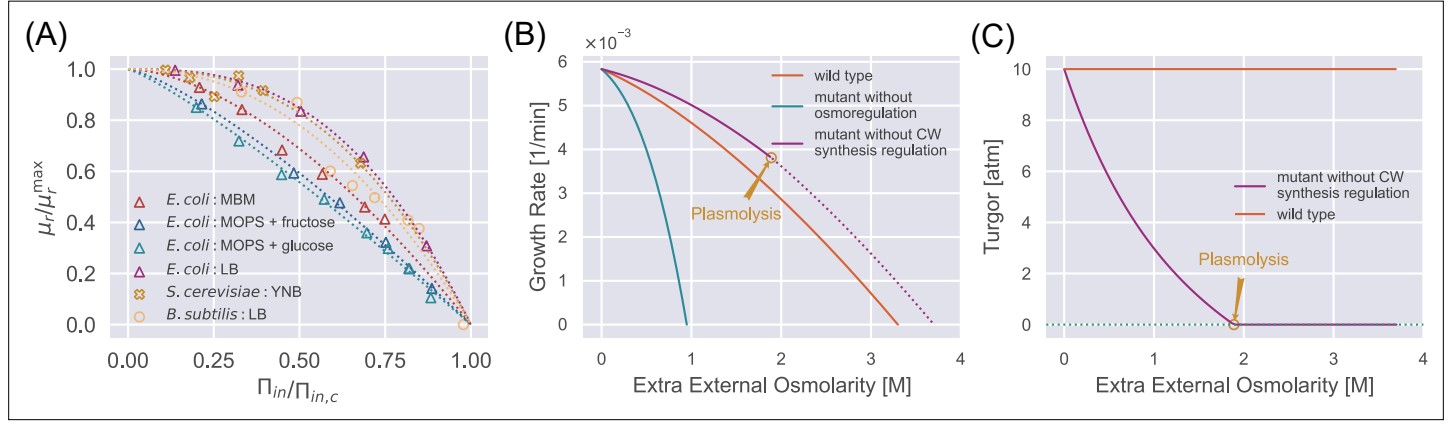

**Figure 2.** Steady-state properties under a constant external osmolarity. (**A**) Normalized growth rate vs. normalized internal osmotic pressure of different species under various culture media. The experiment data (scatter markers) are fitted by our theoretical prediction *Equation 10b*. The data of *E. coli* are from *Cayley et al., 1991*; *Dai et al., 2018*; *Rojas et al., 2014*, the data of *B. subtilis* is from *Rojas et al., 2017*, and the data of *S. cerevisiae* is from our own experiments, where sorbitol is added to increase the external osmolarity. (**B**) Growth curves of wild-type (WT) cells, mutant cells without osmoregulation ($H_a = 0$), and mutant cells without cell-wall synthesis regulation ($H_{cw} = 0$). The dotted line indicates the region where plasmolysis occurs for the mutant cells with $H_{cw} = 0$. (**C**) Mutant cells without cell-wall synthesis regulation cannot maintain a stable turgor pressure in a hypertonic environment, while WT cells can maintain a constant turgor pressure. The mutant cells reach plasmolysis at a threshold of external osmolarity. In (**B**) and (**C**), the parameters for WT cells are chosen as the values for *S. pombe*, and the mutant values are set such that they have the same growth rate as the WT cells in the reference medium (*Appendix 1—table 2*).

The online version of this article includes the following source data for figure 2:

**Source data 1.** Data tables extracted from Figure 1 and Table 1 of *Cayley et al., 1991*.

**Source data 2.** Data tables extracted from Figure 1A of *Dai et al., 2018*.

**Source data 3.** Data tables extracted from Figure 1A of *Rojas et al., 2014*.

**Source data 4.** Data tables extracted from Figure 3D of *Rojas et al., 2017*.

The right-hand side of *Equation 10a* is a constant independent of external osmolarity (see its detailed expression in Section C of Appendix 1). In deriving *Equation 10b*, we have replaced $\rho_p$ by $\Pi_{in}$ in *Equation 8* using *Equation 10a* with the critical internal osmotic pressure $\Pi_{in,c}$ proportional to $\rho_c$. Intriguingly, the relationship between the normalized growth rate ($\mu_r/\mu_r^{\max}$) and the normalized cytoplasmic osmotic pressure ($\Pi_{in}/\Pi_{in,c}$), which we refer to as the growth curve in the following, has only one parameter $H_r/(H_a + 1)$. Therefore, the growth curves of different organisms can be unified by a single formula, *Equation 10b*, and different organisms may have different values of $H_r/(H_a + 1)$. Furthermore, *Equation 10b* predicts a critical external osmolarity $\Pi_{out,c} = \Pi_{in,c} - \sigma_c$, beyond which cell growth is completely inhibited.

We test the validity of *Equation 10b* by fitting it to the experimental growth curves (*Figure 2A*). To do this, we calculate the internal osmotic pressure using *Equation 9* given the values of the external osmotic pressure and the turgor pressure (*Table 1*). Intriguingly, the growth curves of multiple species can be well fitted by *Equation 10b*, from which we infer the parameters $H_r/(H_a + 1)$ and $\Pi_{in,c}$ (*Table 1*). We find that budding yeast cells exhibit notable resilience to high external osmolarities: their $\Pi_{in,c}$ value is higher than those of Gram-positive bacteria, *B. subtilis*, and Gram-negative bacteria, *E. coli*. Further, budding yeast cells demonstrate a higher value of $H_r/(H_a + 1)$, indicating a reduced susceptibility to growth rate reduction when exposed to mild increases in the external osmolarity. Meanwhile, the osmoadaptation capability of *E. coli* depends on the growth media, presumably arising from variations in metabolic fluxes and gene expressions (*Cayley et al., 1991*; *Dai et al., 2018*; *Rojas et al., 2014*).

To further reveal the functions of biological regulations, we study the steady-state properties of mutant cells in which either osmoregulation or cell-wall synthesis regulation is depleted. For mutant cells without osmoregulation, $H_a = 0$ in *Equation 4*. In this case, the fraction of osmolyte-producing protein is constant with time, i.e., $\phi_a = \chi_a^{\max}$. Comparing the dynamics of osmolyte and total protein mass, $\dot{N}_a = k_a \phi_a m_p$ and $\dot{m}_p = k_r \phi_r m_p$, one finds that the ratio of the number of osmolyte molecules and the total protein mass remains constant over time, irrespective of variations in external osmolarity (see the detailed derivation in Section C of Appendix 1). As the external osmolarity increases, the protein density of mutant cells quickly reaches the critical value $\rho_c$ according to *Equation 10a* with $H_a = 0$. Therefore, the steady-state growth curve of the mutant cells terminates at an external osmolarity much smaller than wild-type (WT) cells (*Figure 2B*), in agreement with previous experiments (*Brewster et al., 1993*).

For mutant cells without the cell-wall synthesis regulation, $H_{cw} = 0$; therefore, the cell-wall synthesis efficiency $\eta_{cw}$ equals 1 independent of time. Thus, the growth rate of the relaxed cell-wall volume is always equal to the growth rate of total protein mass (*Equation 6* and *Equation 7*). Interestingly, in this case, the turgor pressure at steady states decreases with the increase of external osmolarity (*Figure 2C* and see the detailed proof in Section C of Appendix 1). The decreased turgor pressure lowers the internal osmotic pressure given the same $\Pi_{out}$ according to *Equation 9*, leading to a lower protein density of mutant cells than WT cells according to *Equation 10a*. Therefore, mutant cells grow faster than WT cells under the same external osmolarity (*Figure 2B*). Nevertheless, the mutant cells are prone to plasmolysis at a threshold external osmolarity where the WT cells can maintain constant turgor pressure (see the vertical line in *Figure 2C* around 2 M extra external osmolarity). Reduced turgor pressure is detrimental to multiple biological processes, e.g., cytokinesis in fission yeast requires the participation of turgor pressure (*Proctor et al., 2012*).

To summarize, osmoregulation allows cells to grow in a wide range of external osmolarity conditions with a mild change in protein density. The cell-wall synthesis regulation allows cells to maintain a stable turgor pressure and avoid plasmolysis. Both regulatory mechanisms expand the range of external osmolarities that cells can adapt to.

## Transient dynamics after a constant osmotic shock

Next, we study the dynamical behaviors of cellular properties in response to a constant osmotic shock: the external osmolarity changes abruptly and keeps its value for an infinitely long time. Intriguingly, we find that the dynamics of osmoresponse can be split into shock and adaptation periods (see insets of *Figure 3C and D*). The immediate water flow due to osmotic imbalance occurs in the shock period, during which the mass and osmolyte productions are negligible. Therefore, the ratio of the internal osmotic pressure and the protein density is invariant right before and after a shock period:

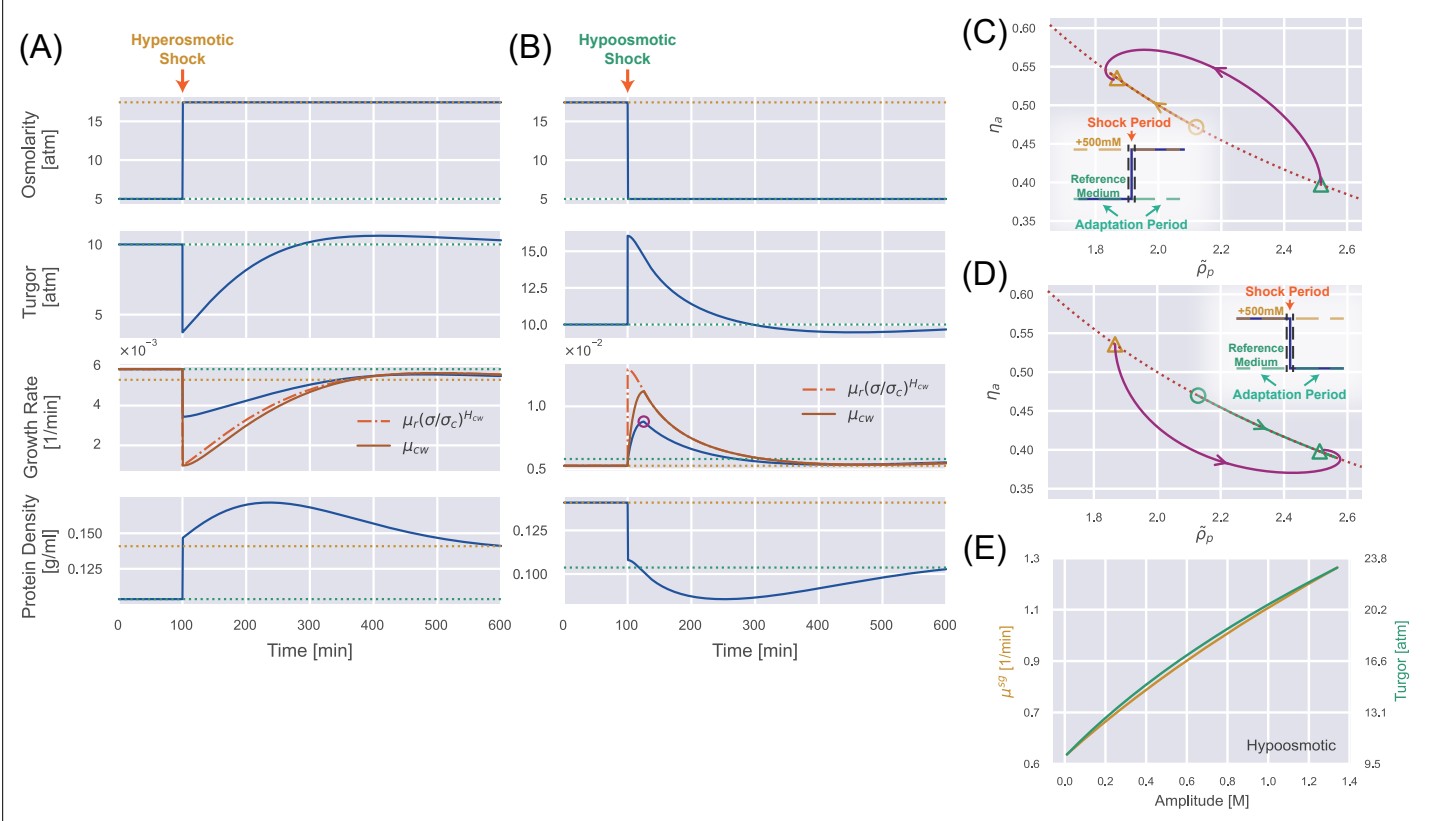

**Figure 3.** Transient dynamics after a constant osmotic shock. (**A**) Numerical simulations of cells undergoing a constant 500 mM hyperosmotic shock. The dotted lines represent the steady-state values for the reference growth medium (green) and the medium after perturbation (yellow). (**B**) Numerical simulations of cells undergoing a constant 500 mM hypoosmotic shock. The purple circle in the third panel marks the growth rate peak during the supergrowth phase. (**C**) The dynamics of the internal state of a cell characterized by $(\tilde{\rho}_p, \eta_a)$. The dotted curve represents the constraint on the steady-state solution $\tilde{\rho}_p, \eta_a = 1$, and the solid trajectory is from numerical simulations. The triangles indicate the steady-state solution before the perturbation and the steady-state solution after the perturbation for a long enough time. The yellow open circle represents the immediate steady-state solution after applying the hyperosmotic shock. (**D**) The same analysis as (**C**) but for a constant 500 mM hypoosmotic shock. (**E**) The growth rate peak in the supergrowth phase (yellow) and the immediate value of turgor pressure after the hypoosmotic shock $\sigma f$ (green) vs. the amplitude of the hypoosmotic shock.

The online version of this article includes the following video and figure supplement(s) for figure 3:

**Figure supplement 1.** Dynamics of other variables under a constant 500 mM hypoosmotic shock.

**Figure 3—video 1.** The trajectory in the internal state space for a wide-type cell during a 500 mM hyperosmotic shock.

https://elifesciences.org/articles/102858/figures#fig3video1

**Figure 3—video 2.** The trajectory in the internal state space for a wide-type cell during a 500 mM hypoosmotic shock.

https://elifesciences.org/articles/102858/figures#fig3video2

$\Pi_{in}^i/\rho_p^i = \Pi_{in}^f/\rho_p^f$, where the upper index $i$ ($f$) means the state right before (after) the shock period. Given this condition, we introduce the normalized protein density $\tilde{\rho}_p$ as

$$\tilde{\rho}_p = \frac{\rho_p}{\bar{\rho}_p}, \tag{11}$$

where the normalization factor $\bar{\rho}_p \propto \Pi_{in}$ (see its detailed expression in Materials and methods) so that $\tilde{\rho}_p$ changes continuously across the shock period. Interestingly, we find that osmoresponse is governed by a two-dimensional dynamical system composed of $\tilde{\rho}_p$ and $\eta_a \equiv \phi_a/\chi_a^{\max}$ (Materials and methods):

$$\frac{\dot{\tilde{\rho}}_p}{\tilde{\rho}_p} = \mu_r^{\max} \eta_r \left(1 - \tilde{\rho}_p \eta_a\right). \tag{12a}$$

$$\dot{\eta}_a = \mu_r^{\max} \eta_r \left[ \left( \frac{\tilde{\rho}_p}{\tilde{\rho}_c} \right)^{H_a} - \eta_a \right], \tag{12b}$$

Here, $\tilde{\rho}_c = \rho_c/\bar{\rho}_p$ is the normalized critical protein density, and $\eta_a$ denotes the efficiency of osmo-regulation. From the above equations, it is clear that the timescale of osmoregulation is set by the doubling time: it takes about the doubling time for the protein density and the fraction of osmolyte-producing protein to adapt to the new steady-state values. For walled cells, $\tilde{\rho}_c$ and $\bar{\rho}_p$ depend on time since $\Pi_{in} = \Pi_{out} + \sigma$ and the turgor pressure $\sigma$ is time-dependent during osmoresponse processes (*Figure 3A and B*). For unwalled cells, such as mammalian cells and microbial cells with cell walls removed (i.e. protoplasts), $\tilde{\rho}_c$ is constant in a fixed environment (see detailed discussion on the transient dynamics of unwalled cells in Section D of Appendix 1).

Upon a constant hyperosmotic shock, the immediate water efflux leads to an instantaneous drop in turgor pressure and a rise in protein density (*Figure 3A*). The internal state of the cell, $(\tilde{\rho}_p, \eta_a)$, evolves toward the new equilibrium point, $(\tilde{\rho}_c^{H_a/(H_a+1)}, \tilde{\rho}_c^{-H_a/(H_a+1)})$. One should note that the equilibrium point is time-dependent initially but eventually becomes fixed as the turgor pressure relaxes to the steady-state value (*Figure 3C* and *Figure 3—video 1*). Interestingly, the protein density increases initially and then decreases after the shock (*Figure 3A*). The decrease in protein density is because of the osmoregulation process, which is set by the doubling time (*Equation 12a* and *Equation 12b*). Meanwhile, we find that the initial increase of protein density is because of the suppressed growth of the relaxed cell-wall volume due to the low turgor pressure. Indeed, for unwalled cells, the protein density $\rho_p$ decreases immediately after the shock (*Appendix 1—figure 2B*). We note that the growth rate approaches the new steady-state value non-monotonically (*Figure 3A*) because of the spiral trajectory in the space of the internal state (*Figure 3C*), consistent with experimental observations from *Rojas et al., 2014*.

The phenomena are essentially the opposite for a constant hypoosmotic shock (*Figure 3B and D*, *Figure 3—video 2*). However, we find extremely fast cell growth after the hypoosmotic shock, with a growth rate peak occurring about 25 min after applying the shock, which we call the super-growth phase (*Knapp et al., 2019*). One should note that 25 min is much shorter than the doubling time (about 2 hr) but comparable to the timescale of cell-wall synthesis regulation, which we set as $\tau_{cw}^+ = 12.5$ min in the simulations in *Figure 3* (we will explain why we choose $\tau_{cw}^+ = 12.5$ min in the next section). Furthermore, applying a hypoosmotic shock to an unwalled cell does not induce a significant supergrowth phase compared with walled cells (*Appendix 1—figure 2D*).

We propose that supergrowth comes from the high turgor pressure caused by the hypoosmotic shock, which leads to fast cell-wall synthesis according to *Equation 7*. Rapid insertion of materials into the cell wall relaxes the turgor pressure and allows the cells to grow faster (*Equation 2* and *Equation 3*). This idea is consistent with the observation that the growth rate and the growth rate of the relaxed cell-wall volume $\mu_{cw}$ reach their peaks simultaneously (*Figure 3B*). This observation also suggests that the timescale of supergrowth, i.e., the timing of growth rate peak, is set by the times-cale of cell-wall synthesis regulation ($\tau_{cw}^+$ in *Equation 7*). Notably, in the initial stage of the adaptation period, $\mu_{cw}$ approaches its target from below and reaches its target value at the growth rate peak (i.e. $\mu_r(\sigma/\sigma_c)^{H_{cw}}$) (the third panel of *Figure 3B*), after which $\mu_{cw}$ sticks to its target value and decreases accordingly because of the short relaxation time $\tau_{cw}^-$ (*Equation 7*). For comparison, we also show $\mu_{cw}$ and $\mu_r(\sigma/\sigma_c)^{H_{cw}}$ for the hyperosmotic shock in the third panel of *Figure 3A*. A detailed proof of the conditions for supergrowth, including the necessity of a cell wall and the regulation of cell-wall synthesis, is provided in Section E of Appendix 1.

Following the discussion above, we obtain an analytical expression of the growth rate peak after a hypoosmotic shock (see the detailed derivations in Section F of Appendix 1)

$$\mu^{sg} = \mu_r \left\{ 1 + \frac{f}{f + \dfrac{\Pi_{in}}{\sigma + G}} \times \left[ \left( \frac{\sigma}{\sigma_c} \right)^{H_{cw}} - 1 \right] \right\}. \tag{13}$$

Here, all the variables on the right side are at the growth rate peak. Because the timescale of the osmoresponse process, which is around hours (*Figure 3B*), is much longer than the timescale of the supergrowth phase, which is about 20 min for *S. pombe* (*Knapp et al., 2019*), the turgor pressure at the growth rate peak can be well approximated by its immediate value after the shock. Therefore,

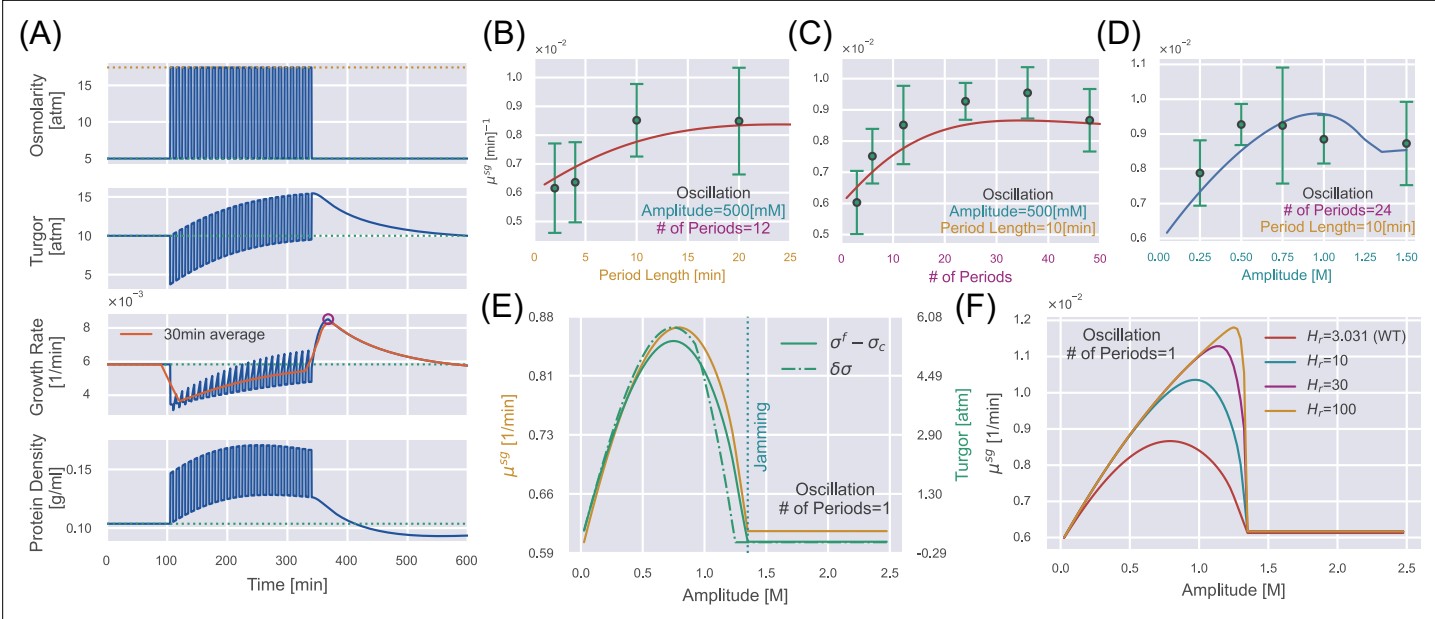

**Figure 4.** Supergrowth phenomena under osmotic oscillation. (**A**) Numerical simulations of wild-type (WT) *S. pombe* undergo 24 cycles of 500 mM osmotic oscillations with a 10 min period. We show a 30 min window average in the third panel of growth rate. (**B–D**) Quantitative agreement between simulations and experiments for the growth rate peak $\mu^{sg}$ vs. different oscillation parameters, including (**B**) amplitude, (**C**) period length, and (**D**) number of periods. The red lines in (**B, C**) are predictions, and the blue line in (**D**) is fitting from which we infer the values of $H_{cw}$ and $\tau_{cw}^+$. Green dots with error bars are experimental data from *Knapp et al., 2019*. (**E**) In the case of osmotic oscillation with a single period, the hyperosmotic period persists for 120 min before reverting to the reference medium. The vertical dotted blue line represents the minimal amplitude to induce cytoplasm jamming during the hyperosmotic period. The excess turgor pressure $\sigma^f - \sigma_c$ upon exiting the hyperosmotic period is approximately equal to the recovered turgor pressure $\delta\sigma$ during the hyperosmotic period. (**F**) The growth rate peak $\mu^{sg}$ at different $H_r$ vs. the amplitude of a single oscillation. $H_r = 3.031$ is the value of the WT *S. pombe*. Parameters of WT *S. pombe* are used in this figure unless otherwise mentioned (*Table 1*).

The online version of this article includes the following source data and figure supplement(s) for figure 4:

**Source data 1.** Data tables extracted from Figure 4A–C of *Knapp et al., 2019*.

**Figure supplement 1.** Dynamics of other variables under 24 cycles of 500 mM osmotic oscillation.

**Figure supplement 2.** The non-monotonic relationship between $\mu^{sg}$ and amplitude.

the growth rate peak must increase as the amplitude of the hypoosmotic shock increases, which we confirm numerically in *Figure 3E*.

## Comparison with experiments: supergrowth phenomena after osmotic oscillation

Next, we quantitatively compare our theoretical predictions regarding the supergrowth phase with experimental data. *Knapp et al., 2019*, applied an osmotic oscillation to fission yeast *S. pombe* during which the external osmolarity alternated between two values. They found cell growth was almost inhibited during the perturbation, while the protein and dry-mass densities increased. Surprisingly, cells grew unusually fast after the osmotic oscillation was removed and reached their maximum growth rate about 20 min after the end of the osmotic oscillation. The maximum growth rate can be twice the growth rate in the reference growth medium, and the elevation in growth rate can persist for two to three cell cycles. These observations are very similar to our results for a constant hypoosmotic shock (*Figure 3B*).

To test if our osmoresponse model captures the supergrowth phase for a periodic perturbation, we simulate WT *S. pombe* cells with the same protocols as the experiments (see details of simulations in Materials and methods). Intriguingly, our model successfully recapitulates the supergrowth phase and the gradually increasing protein density and dry-mass density during the perturbation (*Figure 4*). We confirm that cell-wall synthesis regulation is crucial for the emergence of the supergrowth phase since unwalled cells do not exhibit supergrowth after periodic perturbation (*Appendix 1—figure 3*).

Interestingly, we find that an infinitely long periodic osmotic shock can be equivalently mapped to a constant osmotic shock (see the detailed discussions and proof in Section D of Appendix 1), which means that they have the same time-averaged growth rate and protein density in the steady states (*Appendix 1—figure 2F*).

*Knapp et al., 2019*, measured the growth rate peaks vs. three different parameters of the osmotic oscillations: amplitude, period length, and number of periods. We first fit the growth rate peaks vs. the amplitudes (*Figure 4D*), from which we obtain $H_{cw} = 1.7$, the sensitivity of the cell-wall synthesis efficiency to turgor pressure (*Equation 7*), and $\tau_{cw}^+ = 12.5$ min, the timescale in the upregulation of cell-wall synthesis efficiency (which is why we set $\tau_{cw}^+ = 12.5$ min in the previous section). Other model parameters are inferred from independent steady-state measurements, and we set the timescale in the downregulation of cell-wall synthesis efficiency as $\tau_{cw}^- = 0.1$ min for simplicity (*Table 1*). We next fix the values of $H_{cw}$ and $\tau_{cw}^+$ and plot the predicted growth rate peaks vs. the period length (*Figure 4B*) and number of periods (*Figure 4C*). As a strong support of our model, our predictions quantitatively match the experimental data without any further fitting.

Two interesting features of the curve $\mu^{sg}$ vs. amplitude catch our attention: the non-monotonic behavior and the kink point at which the derivative is discontinuous (*Figure 4D*), which are conserved regardless of the number of periods (*Figure 4—figure supplement 2*). Therefore, we study the case of a single oscillation for simplicity, which is equivalent to a hyperosmotic shock of finite duration. For a mild hyperosmotic shock, during the period of hyperosmotic shock, the turgor pressure has almost recovered to the steady-state value $\sigma_c$ (*Figure 3A*). Therefore, switching from a long hyperosmotic period to the reference growth medium is equivalent to a constant hypoosmotic shock, where we have shown that the growth rate peak increases with the amplitude (*Figure 3E*). However, the crowding effect becomes more pronounced as the amplitude increases. Beyond the critical amplitude at the kink point, the cytoplasm is completely jammed during the hyperosmotic shock such that the cell states are precisely the same before and after the hyperosmotic shock, which means no supergrowth phase beyond this critical amplitude. Therefore, the curve $\mu^{sg}$ vs. amplitude must be non-monotonic (*Figure 4E*). Notably, for a very large $H_r$, cells can feel the crowding effect only when the cytoplasm is close enough to the critical protein density, shown as the abrupt decline of $\mu^{sg}$ (*Figure 4F*).

Finally, we remark that the significance of supergrowth is intimately related to the amount of recovered turgor pressure during the hyperosmotic shock $\delta\sigma$. We prove that the overshoot of turgor pressure after the removal of hyperosmotic shock ($\sigma^f - \sigma_c$), which sets the growth rate peak, is mainly set by the recovered turgor pressure during the hyperosmotic shock (see the detailed discussions in Section G of Appendix 1). Indeed, $\mu^{sg}$, $\sigma^f - \sigma_c$, and $\delta\sigma$ are highly correlated as we change the amplitude (*Figure 4E*).

## Discussion

This study presents a theory of microbial osmoresponses based on a physical foundation and simplified biological regulation strategies. Our theory captures the steady-state properties of constant turgor pressure and reduced growth rate with increasing external osmolarity. We remark that the growth rate reduction is due to the loss of free water and subsequent intracellular crowding as the external osmolarity increases. In particular, we predict a critical external osmolarity above which cell growth is completely inhibited and a universal relationship between the normalized growth rate and the normalized internal osmotic pressure, fitting the data of bacteria and yeast. We also demonstrate the biological functions of osmoregulation and cell-wall synthesis regulation. Cells defective in osmoregulation cannot grow even if the external osmolarity is only mildly higher than the reference value. Cells defective in cell-wall synthesis regulation cannot maintain turgor pressure as the external osmolarity increases, even though they grow faster than WT cells (*Figure 2B*), which will be a strong support of our theory if confirmed by experiments.

Regarding dynamic behaviors, our model predicts a non-monotonic time dependence of protein density after a constant hyperosmotic shock. We also unveil the supergrowth phase after a hypoosmotic shock, initially discovered in fission yeast after an osmotic oscillation (*Knapp et al., 2019*). As a strong support of our theory, the predicted growth rate peaks quantitatively agree with the experimental data without additional fitting. We demonstrate the critical role of cell-wall synthesis regulation in the supergrowth phenomenon (Section E of Appendix 1). *Knapp et al., 2019*, observed the rapid repolarization of the cell-wall glucan synthase Bgs4 to the cell tip following the removal of

osmotic oscillations in fission yeast, in agreement with the dynamics of the cell-wall synthesis efficiency predicted from our model (compare *Figure 4—figure supplement 1* in this work and Figure S4H in *Knapp et al., 2019*). To test our theory, we propose applying a hyperosmotic shock with a finite duration and measuring the growth rate after removing the hyperosmotic shock. We predict that the growth rate peak during the supergrowth phase is a non-monotonic function of shock amplitude, initially rising because of the increased excess turgor pressure and later declining because the protein density reaches the critical value $\rho_c$ during the shock (*Figure 4E*).

We remark that our model is intrinsically a coarse-grained model with many molecular details regarding gene expression regulation neglected, which allows us to gain more analytical insights. *Shen et al., 2023*, studied the responses to osmotic stress in glucose-limited environments and found that cells exhibited stronger osmotic gene expression response under glucose-limited conditions than under glucose-rich conditions. Using a computational model based on molecular mechanisms combined with experimental measurements, the authors demonstrated that in a glucose-limited environment, glycolysis intermediates were limited, which required cells to express more glycerol-production enzymes for stress adaptation. In the current version of our model, we do not account for the interaction between cell growth and osmolyte production; instead, we assume a constant fraction of ribosomes dedicated to translating ribosomal proteins. Our model can be further generalized to include the more complex interactions, including the coupling between biomass and osmolyte production, e.g., by allowing the fraction of ribosomes translating ribosomal proteins ($\chi_r$) to depend on the translation strategy of the osmolyte-producing enzyme ($\chi_a$).

*Rojas et al., 2014*, showed that the expansion of *E. coli* cell wall is not directly regulated by turgor pressure, and this scenario is also included in our model as the case of $H_{cw} = 0$. According to our model, the supergrowth phase is absent if $H_{cw} = 0$ (*Appendix 1—figure 8*), consistent with the absence of a growth rate peak after a hypoosmotic shock in the experiments of *E. coli* (*Rojas et al., 2014*). Meanwhile, our predictions are consistent with the growth rate peak after a hypoosmotic shock observed for *B. subtilis* (*Rojas et al., 2017*).

We remark on several limitations of our current coarse-grained model. First, the high membrane tension that inhibits transmembrane flux of peptidoglycan precursors, leading to a growth inhibition before the supergrowth peak (*Rojas et al., 2017*), is beyond our model. Second, in our current framework, osmoregulation and cell-wall synthesis regulation rely on the instantaneous cellular states. However, microorganisms can exhibit memory effects to external stimuli by adapting to their temporal order of appearance (*Mitchell et al., 2009*). Notably, in the osmoregulation of yeast, a short-term memory, facilitated by posttranslational regulation of the trehalose metabolism pathway, and a long-term memory, orchestrated by transcription factors and mRNP granules, have been identified by *Jiang et al., 2020*. Besides, our model does not account for the role of osmolyte export in osmoregulation (*Tamás et al., 1999*) and the interaction between biomass and osmolyte production (*Shen et al., 2023*). Extending our model to include more realistic biological processes will be interesting.

In this work, we construct a systems-level and coarse-grained model capable of elucidating the complex processes underlying microbial osmoresponse. By leveraging the separation of timescales associated with mechanical equilibrium, cell-wall synthesis regulation, and osmoregulation, our model facilitates in-depth analytical and numerical analysis of how these various processes interact during cellular adaptation. In particular, we demonstrate the key physiological functions of osmoregulation and cell-wall synthesis regulation. We then apply this model to interpret the unusual phenomenon of supergrowth observed in fission yeast. This application addresses an essential challenge in experimental studies: exclusive knockout experiments can be difficult, and mechanistic interpretations of experimental observations are often lacking. Our theoretical framework offers a valuable tool for understanding such phenomena, contributing to the fundamental knowledge of microbial physiology and developing predictive models for microbial behaviors under osmotic stress.

## Materials and methods
### Details of the osmoresponse model
We define the fractions of osmolyte-producing protein and ribosomal proteins in the total proteome as $\phi_a = m_{p,a}/m_p$ and $\phi_r = m_{p,r}/m_p$, respectively. To model gene expression regulation, we introduce

$\chi_a$ and $\chi_r$ as the fractions of ribosomes translating the osmolyte-producing protein and ribosomal proteins such that

$$\dot{m}_{p,r} = k_r \chi_r m_{p,r} \Rightarrow \dot{\phi}_r = k_r \phi_r (\chi_r - \phi_r), \tag{14}$$

$$\dot{m}_{p,a} = k_r \chi_a m_{p,a} \Rightarrow \dot{\phi}_a = k_r \phi_a (\chi_a - \phi_a), \tag{15}$$

$$\dot{m}_p = k_r m_{p,r} \Rightarrow \mu_r = k_r \phi_r. \tag{16}$$

Here, $k_r$ is proportional to the elongation speed of ribosomes on mRNAs divided by the protein mass of a single ribosome, which is affected by the global crowding effect as $k_r = k_r^{\max} \eta_r$. Here, $\mu_r$ is the growth rate of total protein mass, which is also the growth rate of dry mass and bound volume in our model since they are all proportional. The osmolyte molecules are produced by the osmolyte-producing protein, with the rate given by

$$\dot{N}_a = k_a m_{p,a}, \tag{17}$$

where $k_a = k_a^{\max} \eta_r$ is the osmolyte production rate, including the crowding factor, and $m_{p,a}$ is the mass of osmolyte-producing protein. We summarize the dynamical equations involved in the osmoresponse model:

$$\dot{\rho}_p = (\mu_r - \mu_f)\rho_p \tag{18a}$$

$$\dot{\eta}_a = \mu_r \left[ \left( \frac{\rho_p}{\rho_c} \right)^{H_a} - \eta_a \right] \tag{18b}$$

$$\dot{\Pi}_{in} = k_B T k_a^{\max} \eta_r \phi_a \rho_p - \mu_f \Pi_{in} \tag{18c}$$

$$\dot{\epsilon} = (\mu - \mu_{cw})(\epsilon + 1) \tag{18d}$$

$$\dot{\eta}_{cw} = \frac{1}{\tau_{cw}^{\pm}} \left[ \left( \frac{\sigma}{\sigma_c} \right)^{H_{cw}} - \eta_{cw} \right]. \tag{18e}$$

To describe the osmoregulation process using a two-dimensional dynamical system, we introduce the normalized protein density as

$$\tilde{\rho}_p = \frac{k_B T k_a^{\max} \chi_a^{\max}}{\mu_r^{\max}} \frac{\rho_p}{\Pi_{in}} \equiv \frac{\rho_p}{\bar{\rho}_p}, \tag{19}$$

Combining *Equation 11* and *Equation 18a*, we obtain the dynamical equation for $\tilde{\rho}_p$ as

$$\frac{\dot{\tilde{\rho}}_p}{\tilde{\rho}_p} = \mu_r^{\max} \eta_r \left( 1 - \tilde{\rho}_p \eta_a \right). \tag{20}$$

Using *Equation 15*, we obtain the equation for $\eta_a = \phi_a / \chi_a^{\max}$ as

$$\dot{\eta}_a = \mu_r^{\max} \eta_r \left[ \left( \frac{\tilde{\rho}_p}{\tilde{\rho}_c} \right)^{H_a} - \eta_a \right], \tag{21}$$

where $\tilde{\rho}_c = \rho_c / \bar{\rho}_p$. The unique equilibrium point for the internal state is

$$(\tilde{\rho}_p^{eq}, \eta_a^{eq}) = \left( \tilde{\rho}_c^{\frac{H_a}{H_a + 1}}, \tilde{\rho}_c^{-\frac{H_a}{H_a + 1}} \right). \tag{22}$$

## Details of numerical simulations

We employ the LSODA algorithm with automatic stiffness detection and switching (*Petzold, 1983*), implemented in SciPy (*Virtanen et al., 2020*), to solve *Equation 18a–e*. The parameters used for numerical simulations of walled cells are listed in *Table 1*.

## Acknowledgements

We thank Chunxiong Luo for helpful discussions related to this work. The research was funded by the National Key Research and Development Program of China (2024YFA0919600), the National Natural Science Foundation of China (Grant No. 12474190), and Peking-Tsinghua Center for Life Sciences grants.

## Additional information

### Funding

| Funder | Grant reference number | Author |
| --- | --- | --- |
| National Key Research and Development Program of China | 2024YFA0919600 | Jie Lin |
| National Natural Science Foundation of China | 12474190 | Jie Lin |
| Peking-Tsinghua Center for Life Sciences | | Jie Lin |

The funders had no role in study design, data collection and interpretation, or the decision to submit the work for publication.

### Author contributions

Yiyang Ye, Conceptualization, Software, Formal analysis, Validation, Investigation, Visualization, Methodology, Writing – original draft, Writing – review and editing; Qirun Wang, Resources, Data curation, Investigation; Jie Lin, Conceptualization, Formal analysis, Supervision, Funding acquisition, Writing – original draft, Project administration, Writing – review and editing

### Author ORCIDs

Yiyang Ye ⬤ https://orcid.org/0000-0002-2411-5297
Qirun Wang ⬤ https://orcid.org/0000-0001-8845-1997
Jie Lin ⬤ https://orcid.org/0000-0002-2027-4661

Reviewer #1 (Public review): https://doi.org/10.7554/eLife.102858.3.sa1
Reviewer #2 (Public review): https://doi.org/10.7554/eLife.102858.3.sa2
Author response https://doi.org/10.7554/eLife.102858.3.sa3

## Additional files

### Supplementary files
MDAR checklist

### Data availability

All data analysed during this study are included in the manuscript and supporting files. Source data files have been provided for *Figures 2 and 4*, *Appendix 1—figure 1*. *Figure 2—source data 1–4* contain the experimental data used to fit and validate our theory in panel A of *Figure 2*. *Figure 4—source data 1* contains the experimental data used to fit and validate our theory in panels B to D of *Figure 4*. *Appendix 1—figure 1—source data 1* contains the experimental data used to fit our model parameters in *Appendix 1—figure 1*.

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

# Appendix 1

## A. Conversion between the dry-mass density and the protein density

In the main text, we mainly focus on the density of cytoplasmic proteins, defined as $\rho_p = m_p/V_f$, which is related to the dry-mass density, defined as $\hat{\rho}_d = m_{tot,d}/V_{tot}$. Here, the subscript *tot* means the total mass and cell volume, encompassing regions outside the cytoplasm. Taking yeast as an example, the dry mass in the cell wall contributes to approximately 30% of the total dry mass (**Lesage and Bussey, 2006**), with mannoproteins constituting 40% of the dry mass in the cell wall (**Klis et al., 2002**). Therefore, a non-negligible proportion of proteins resides outside the cytoplasm in microorganisms. For simplicity, we assume that the fraction of protein mass $m_{tot,p}$ in the total dry mass $m_{tot,d}$ remains constant under different external osmolarities for a given species. Also, this constancy holds both inside and outside the cytoplasm such that

$$\frac{m_p}{m_d} = \frac{m_{tot,p}}{m_{tot,d}} = \text{const.} \tag{A1}$$

Further, we assume that the mass fraction of the cytoplasmic proteins in total proteins is constant, from which we infer the ratio between the total bound volume and the cytoplasmic bound volume: $(V_{tot} - V_f)/V_b = m_{tot,p}/m_p$. With these assumptions, $\rho_p$ and $\hat{\rho}_d$ are related as

$$\rho_p = \frac{m_p}{m_d}\left(\frac{m_p}{m_{tot,p}} + f^{-1} - 1\right)\hat{\rho}_d. \tag{A2}$$

Here, $f$ is the fraction of free volume in the cytoplasmic volume, $f = V_f/V$. In **Appendix 1—table 1**, we present the dry-mass fraction of protein $m_p/m_d$ and the cytoplasmic protein mass fraction $m_p/m_{tot,p}$ used in this work. The same parameters are used for *S. pombe* and *S. cerevisiae* for simplicity. Given **Equation A2** and the value of $f$, which we discuss in the next section, we obtain the protein density from the dry-mass density.

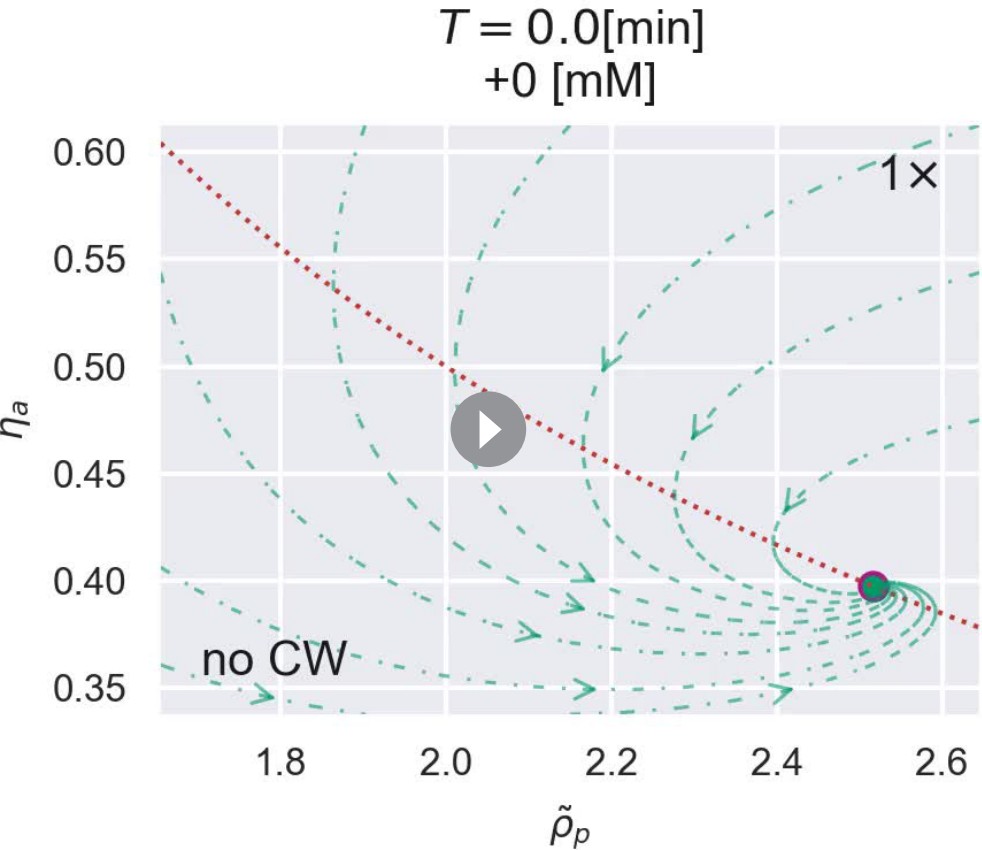

**Appendix 1—video 1.** The trajectory in the internal state space for an unwalled cell during a 500 mM osmotic oscillation with a 10 min period. The video shown here corresponds to the dynamics depicted in *Appendix 1—figure 2E*.

https://elifesciences.org/articles/102858/figures#video1

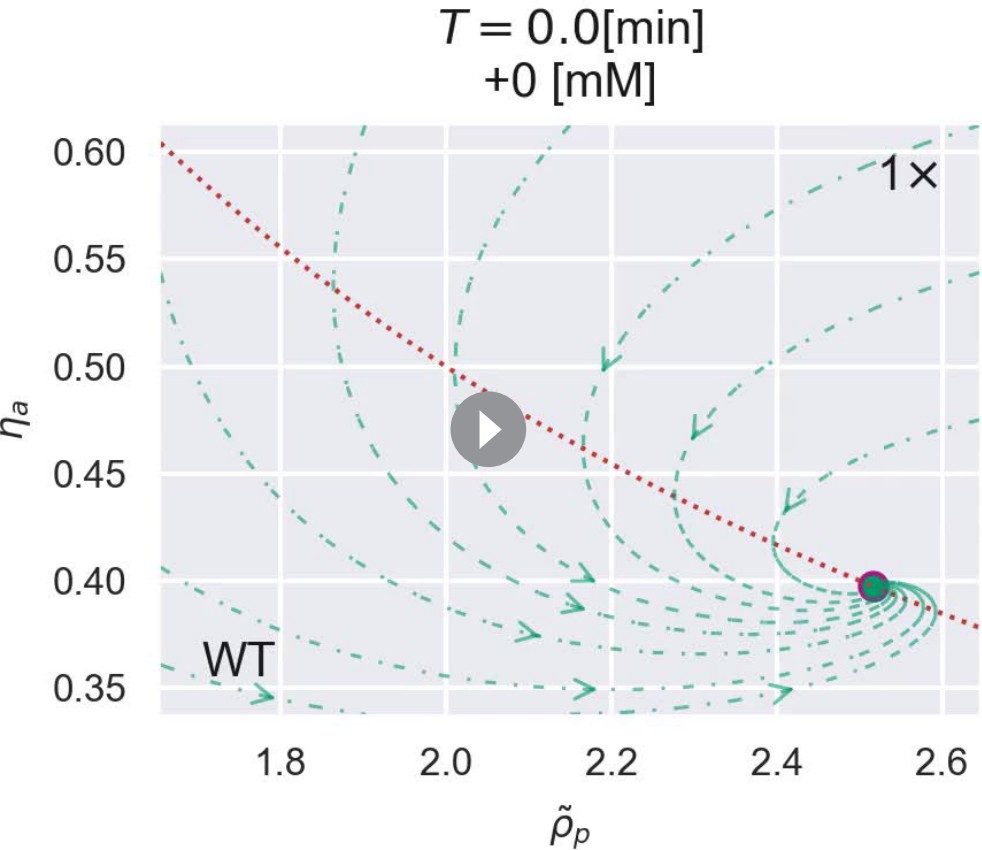

**Appendix 1—video 2.** The trajectory in the internal state space for a wide-type cell during a 500 mM osmotic oscillation with a 10 min period. The video shown here corresponds to the dynamics depicted in **Appendix 1—figure 6**.

https://elifesciences.org/articles/102858/figures#video2

**Appendix 1—table 1.** The fraction of protein mass in the total dry mass, $\frac{m_p}{m_d}$ $\left(= \frac{m_{tot,p}}{m_{tot,d}}\right)$, and the fraction of cytoplasmic protein mass in the total protein mass, $m_p/m_{tot,p}$.

| E. coli | Value | Reference |
|---|---|---|
| $p/m_{tot,p}$ | 0.8 | *Cayley et al., 1991* |
| $m_p/m_d$ | 0.68 | *Cayley et al., 1991* |
| **S. cerevisiae and S. pombe** | **Value** | **Reference** |
| $m_p/m_{tot,p}$ | 0.65 | *Chong et al., 2015* |
| $m_p/m_d$ | 0.4 | *Yamada and Sgarbieri, 2005* |

**Appendix 1—table 2.** Comparison of the model parameters between defective cells and intact cells.
Variables with $*$ are for the reference growth medium.

| Parameters | S. pombe (WT) | S. pombe (No osmoregulation) | S. pombe (No CW synthesis regulation) | S. pombe (No CW) |
|---|---|---|---|---|
| $\Pi_{out}^*$ | 0.2 [Osm] | 0.2 [Osm] | 0.2 [Osm] | 0.2 [Osm]+ 10 [atm] |
| $\sigma^*$ | 10 [atm] | 10 [atm] | 10 [atm] | |
| $\rho_p^*$ | 0.104 [g/ml] | 0.104 [g/ml] | 0.104 [g/ml] | 0.104 [g/ml] |
| $\mu^*$ | 0.35 [1/hr] | 0.35 [1/hr] | 0.35 [1/hr] | 0.35 [1/hr] |
| $f^*$ | 0.788 | 0.788 | 0.788 | 0.788 |
| $\epsilon^*$ | 0.584 | 0.584 | 0.584 | |
| $\alpha$ | 2.60 [ml/g] | 2.60 [ml/g] | 2.60 [ml/g] | 2.60 [ml/g] |
| $G$ | 17.1 [atm] | 17.1 [atm] | 17.1 [atm] | |
| $\Pi_{out,c}$ | 3.5 [Osm] | 1.15 [Osm] | 3.5 [Osm] + 10 [atm] | 3.5 [Osm] + 10 [atm] |
| $k_w$ | 100 [1/(min atm)] | 100 [1/(min atm)] | 100 [1/(min atm)] | 100 [1/(min atm)] |
| $\rho_c$ | 0.267 [g/ml] | 0.267 [g/ml] | 0.267 [g/ml] | 0.267 [g/ml] |
| $\sigma_c$ | 10 [atm] | 10 [atm] | 10 [atm] | |
| $H_r$ | 3.03 | 3.03 | 3.03 | 3.03 |
| $H_a$ | 0.974 | 0 | 0.974 | 0.974 |
| $k_r^{max}\chi_r$ | 0.371 [1/hr] | 0.371 [1/hr] | 0.371 [1/hr] | 0.371 [1/hr] |
| $k_B T k_a^{max}\chi_a^{max}$ | 2.25 [(atm ml)/(g min)] | 0.894 [(atm ml)/(g min)] | 2.25 [(atm ml)/(g min)] | 2.25 [(atm ml)/(g min)] |
| $\tau_{cw}^-$ | 0.1 [min] | 0.1 [min] | | |
| $\tau_{cw}^+$ | 12.5 [min] | 12.5 [min] | | |
| $H_{cw}$ | 1.7 | 1.7 | 0 | |

**Appendix 1—table 3.** A summary of the symbols involved in our model.

| Symbol | Description |
|---|---|
| $V$ | total cytoplasmic volume |
| $V_f$ | free volume: cytoplasmic volume occupied by free water |
| $V_b$ | bound volume: cytoplasmic volume occupied by dry mass and bound water |
| $V_{bw}$ | volume occupied by bound water |
| $V_{bd}$ | dry volume: volume occupied by dry mass |
| $V_{cw}$ | relaxed cell-wall volume |
| $f$ | free volume fraction |
| $\alpha$ | bound volume per total protein mass |
| $\Pi_{in}$ | cytoplasmic osmotic pressure |
| $\Pi_{out}$ | external osmotic pressure |
| $\Pi_{in,c}$ | critical cytoplasmic osmolarity where cell growth arrests |
| $\Pi_{out,c}$ | critical external osmolarity where cell growth arrests |
| $\sigma$ | turgor pressure |

*Appendix 1—table 3 Continued on next page*

*Appendix 1—table 3 Continued*

| Symbol | Description |
| --- | --- |
| $G$ | cell-wall elastic modulus |
| $\epsilon$ | elastic strain of the cell wall |
| $k_w$ | water permeability of the cell membrane |
| $N_a$ | number of osmolyte molecules in cytoplasm |
| $m_p$ | total mass of the proteome |
| $m_{p,a}$ | mass of the osmolyte-producing protein |
| $m_{p,r}$ | mass of the ribosomal protein |
| $\rho_p$ | protein density |
| $\rho_c$ | critical protein density where cell growth arrests |
| $\tilde{\rho}_p$ | normalized protein density |
| $\tilde{\rho}_c$ | normalized critical protein density |
| $\bar{\rho}_p$ | normalization factor in protein density |
| $\phi_a$ | mass fraction of the osmolyte-producing protein in total proteome |
| $\phi_r$ | mass fraction of the ribosomal protein in total proteome |
| $\chi_a$ | fraction of ribosome translating osmolyte-producing protein |
| $\chi_r$ | fraction of ribosome translating ribosomal protein |
| $\chi_a^{\mathrm{max}}$ | largest possible fraction of ribosome translating osmolyte-producing protein |
| $H_a$ | sensitivity of osmoregulation to intracellular crowding |
| $H_{cw}$ | sensitivity of cell-wall synthesis regulation to turgor pressure |
| $\tau_{cw}^{\pm}$ | timescale of up(down)-regulation of cell-wall synthesis regulation |
| $k_a$ | osmolyte production rate |
| $k_a^{\mathrm{max}}$ | maximum osmolyte production rate |
| $k_r$ | ribosomal protein production rate |
| $k_r^{\mathrm{max}}$ | maximum ribosomal protein production rate |
| $\eta_a$ | efficiency of osmoregulation |
| $\eta_{cw}$ | efficiency of cell-wall synthesis regulation |
| $\eta_r$ | crowding factor |
| $\mu$ | growth rate of total volume |
| $\mu_r$ | growth rate of ribosomal protein (dry mass) |
| $\mu_f$ | growth rate of free volume |
| $\mu_{cw}$ | growth rate of relaxed cell-wall volume |
| $\mu^{sg}$ | peak growth rate of total volume during supergrowth phase |

## B. Estimations of the model parameters

In this section, all variables with the superscript $*$ are for the reference growth media we show in *Table 1*.

## 1. The free volume fraction of *S. cerevisiae*

Upon an extreme hyperosmotic shock, the cytoplasm expels all free water such that the expelled volume fraction is equal to the free volume fraction $f$ in the reference growth medium, which is how we compute the free volume fraction for *S. cerevisiae* according to *Miermont et al., 2013*, in *Table 1*.

## 2. The free volume fraction of *S. pombe*

*Molines et al., 2022*, measured the immediate changes in cytoplasmic volume after a constant hyperosmotic shock with varying amplitudes, generating a curve of $V^f/V^*$ vs. $\Delta\Pi_{out}$, where $\Delta\Pi_{out}$ is the extra external osmolarity relative to the reference growth medium. As mentioned in the main text, there is a separation of timescales between the shock periods and the adaptation periods. The number of osmolyte molecules $N_a$ remains conserved during the shock periods. Consequently, the cytoplasmic volume right after the hyperosmotic shock is given by

$$V^f = V_f{}^f + V_b^* = \left[ \frac{\Pi_{in}^*}{\Pi_{in}^f} f^* + (1 - f^*) \right] V^*$$

(A3)

This relationship has been verified for various cell types (*Zhou et al., 2009*; *Ting-Beall et al., 1993*). Moreover, the relaxed cell-wall volume is also conserved during the shock periods. Using *Equation A3*, the turgor pressure after the shock $\sigma^f$ can be related to the turgor pressure of the reference growth medium by

$$\frac{\sigma^f}{\sigma^*} = 1 + \left( 1 + \epsilon^{*-1} \right) \left( \frac{\Pi_{out}^* + \sigma^*}{\Pi_{out}^f + \sigma^f} - 1 \right) f^*.$$

(A4)

We note that the left-hand side of *Equation A4* increases monotonically with $\sigma^f$, while the right-hand side decreases monotonically. Therefore, given the values of $\epsilon^*$, $f^*$, and $\sigma^*$, $\sigma^f$ can be solved uniquely from *Equation A4*. In the case of an extreme hyperosmotic shock (i.e. a very large $\Pi_{out}^f$), no positive solution of $\sigma^f$ can be found, corresponding to plasmolysis. After substituting $\Pi_{in}^f = \Pi_{out}^f + \sigma^f$ into *Equation A3*, $V^f/V^*$ is obtained accordingly. Therefore, we infer the values of $f^*$ for *S. pombe* growing in YE5S media using the value of steady-state turgor pressure $\sigma^*$ in *Table 1* (*Lemière and Chang, 2023*), the value of steady-state cell-wall strain $\epsilon^*$(*Atilgan et al., 2015*), and the data of $V^f/V^*$ vs. $\Delta\Pi_{out}$ (*Appendix 1—figure 1*).

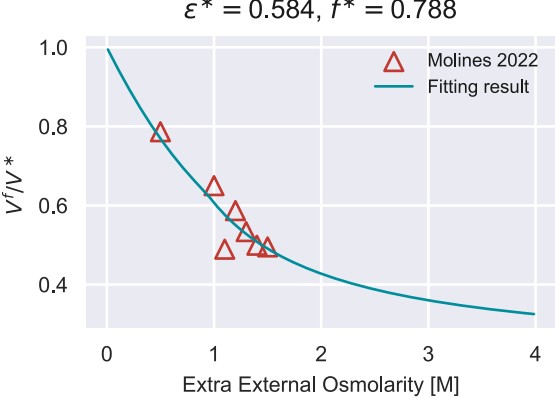

**Appendix 1—figure 1.** Fit of $V^f/V^*$ vs. $\Delta\Pi_{out}$ for *S. pombe* growing in YE5S media, from which we infer the value of $f^*$. The data is extracted from *Molines et al., 2022*.

The online version of this article includes the following source data for appendix 1—figure 1:

**Appendix 1—figure 1—source data 1.** Data tables extracted from Figure S1A of *Molines et al., 2022*.

## 3. Determination of $\rho_c$

*Molines et al., 2022*, studied the changes in intracellular biochemical processes and cytoplasmic diffusion coefficients in response to hyperosmotic shock. The authors found that in acute response to hyperosmotic shocks, the interphase microtubule cytoskeleton appeared to freeze at high external osmolarities. Similar phenomena were found when applying a hyperosmotic shock on *S. cerevisiae* (*Miermont et al., 2013*), where several intracellular signaling cascades, including the stress-response pathways, are significantly slowed down. In our model, the critical external osmolarity $\Pi_{out,c}$ that freezes the cytoplasm happens when $\rho_p = \rho_c$. Therefore, $\rho_c$ in *Table 1* can be derived as follows:

$$\frac{\rho_c}{\rho_p^*} = \frac{\Pi_{out}^* + \sigma^*}{\Pi_{out,c}}. \tag{A5}$$

In the above equation, we have used the fact that the number of osmolyte molecules is conserved during the shock periods and the condition that plasmolysis occurs at the critical external osmolarity such that the turgor pressure is zero under the relevant parameters of *S. cerevisiae* and *S. pombe* (*Table 1*).

## 4. Determination of $\alpha$

We define $\alpha$ as the proportionality coefficient between the bound volume $V_b$ and the cytoplasmic protein mass $m_p$, $\alpha = V_b/m_p$. Thus, one can find $\alpha$ as a function of $f^*$ and $\rho_p^*$ through the following equation:

$$\alpha \rho_p^* = \frac{1}{f^*} - 1, \tag{A6}$$

which is how we compute $\alpha$ for *S. cerevisiae* and *S. pombe* in *Table 1*. We remark that the above equation is also valid for nonsteady states.

We compute $\alpha$ for *E. coli* alternatively. *Scott Cayley et al., 2000*, applied a severe osmotic shock to *E. coli*, squeezing out all of the free water $V_f$, and the remaining volume of bound water per dry mass $V_{bw}/m_{tot,d}$ was computed. Consequently, $\alpha$ can be deduced from the formula:

$$\alpha = \frac{V_{bd}}{m_p} + \frac{V_{bw}}{m_p} = \left( \rho_0^{-1} + \frac{V_{bw}}{m_{tot,d}} \frac{m_{tot,d}}{m_d} \right) \frac{m_d}{m_p}. \tag{A7}$$

Here, $\rho_0 = 1.59$ g/ml represents the density of the dry mass, a constant independent of the external osmolarity (*Cayley et al., 1991*).

## C. Steady-state properties of mutant cells

### 1. Mutant cells without osmoregulation

In steady states, the osmoregulation efficiency is fully determined by the cytoplasmic protein density, $\eta_a = \phi_a/\chi_a^{\max} = (\rho_p/\rho_c)^{H_a}$. After substituting $\dot{\Pi}_{in} = 0$ and $\mu_r = \mu_f$ into *Equation 18c*, we reach the constant as shown in *Equation 10a*:

$$\frac{\Pi_{in}}{\rho_p^{H_a+1}} = \frac{k_B T k_a^{\max} \chi_a^{\max}}{\rho_c^{H_a} \mu_r^{\max}}. \tag{A8}$$

In the case of the osmoregulation-defective cells, $H_a = 0$, and the osmoregulation efficiency remains constant ($\eta_a = 1$), irrespective of variation in the cytoplasmic protein density $\rho_p$. Considering a steady state denoted by $i$, according to *Equation A8*, the ratio between the internal osmotic pressure and the protein density is

$$\frac{\Pi_{in}^i}{\rho_p^i} = \frac{k_B T k_a^{\max} \chi_a^{\max}}{k_r^{\max} \chi_r} \tag{A9}$$

Combining *Equation 18a*, *Equation 18c*, and *Equation A9*, we find that the ratio is time-independent even in transient states because

$$\frac{\mathrm{d}}{\mathrm{d}t}\left(\frac{\Pi_{in}}{\rho_p}\right) = \mu_r\left(\frac{\Pi_{in}^i}{\rho_p^i} - \frac{\Pi_{in}}{\rho_p}\right) = 0. \tag{A10}$$

The second equality arises from the argument that at the initial state when $t = 0$, $\Pi_{in}/\rho_p = \Pi_{in}^i/\rho_p^i$. Therefore, $\Pi_{in}/\rho_p = \Pi_{in}^i/\rho_p^i$ establishes during the entire osmoresponse process. Intuitively, this constancy is due to the alignment between the increasing rates of the number of osmolyte molecules and the total protein mass,

$$\dot{N}_a = k_a^{\max}\chi_a^{\max}\eta_r m_p, \tag{A11a}$$

$$\dot{m}_p = k_r^{\max}\chi_r\eta_r m_p. \tag{A11b}$$

The ratio of the two increasing rates is always constant and independent of time.

## 2. Mutant cells without cell-wall synthesis regulation

For mutant cells with cell-wall synthesis regulation knocked out, the parameter $\mu_{cw}$ is no longer subject to regulation by turgor pressure. This condition is equivalent to setting $H_{cw} = 0$ and $\eta_{cw} = 1$ in our model. Consequently, the cell-wall and dry-mass productions proceed at the same pace, $\mu_{cw} = \mu_r$. Combing *Equation 18a* and *Equation 18d*, we find the following conserved quantity:

$$\frac{\mathrm{d}}{\mathrm{d}t}\ln\left(\frac{\epsilon + 1}{\alpha + \rho_p^{-1}}\right) = \frac{\dot{\epsilon}}{\epsilon + 1} + \frac{f\dot{\rho}_p}{\rho_p} = 0. \tag{A12}$$

The relationship between $\alpha$ and $f$ (*Equation A6*) is utilized in the above derivation. Imagine the external osmotic pressure increases quasistatically, the protein density increases accordingly (*Equation A8*), leading to decreased turgor pressure according to *Equation A12*; *Figure 2C*.

The emergence of this conserved quantity can also be seen from the definition of the elastic strain $1 + \epsilon = V/V_{cw}$ and $m_p/V = m_p/(V_b + V_f) = 1/(\alpha + \rho_p^{-1})$, which gives rise to the following relationship:

$$\frac{m_p}{V_{cw}} = \frac{\epsilon + 1}{\alpha + \rho_p^{-1}}. \tag{A13}$$

Since $m_p/V_{cw}$ is constant because $\mu_r = \mu_{cw}$, we obtain *Equation A12*.

## D. Dynamics of unwalled cells
### 1. Constant osmotic shock
The dynamics of unwalled cells (as described in *Equation 18a* with $\sigma = 0$) consists of three independent variables: $\Pi_{in}$, $\rho_p$, and $\eta_a$. Without turgor pressure, $\Pi_{in} = \Pi_{out}$ holds throughout the adaptation periods. Therefore, the osmoregulation process of unwalled cells can be completely represented by the 2D trajectory of internal state $(\tilde{\rho}_p, \eta_a)$, where $\tilde{\rho}_p$ is the normalized protein density. Since the normalization factor introduced in *Equation 19* is solely dependent on the external osmolarity

$$\bar{\rho}_p = \frac{\mu_r^{\max}}{k_B T k_a^{\max}\chi_a^{\max}}\Pi_{out}, \tag{A14}$$

the steady state defined by $\tilde{\rho}_c$ is time-independent for unwalled cells.

In *Appendix 1—figure 2A and C*, we present the osmoresponse processes of unwalled cells to hyper/hypoosmotic shocks in the two-dimensional space of the internal state. Due to the spiral nature of these trajectories, the growth rates converge to new steady-state values in a non-monotonic manner (*Appendix 1—figure 2B and D*), where the characteristic timescale of the adaptation process is set by the doubling time. Comparing the response of walled cells in *Figure 3A and B*, we find that the presence of cell-wall buffers the drastic changes in protein density induced by osmotic shocks. In particular, after a constant hypoosmotic shock, the growth rate of unwalled cells decreases immediately without supergrowth, unlike the walled cells as shown in *Figure 3B*.

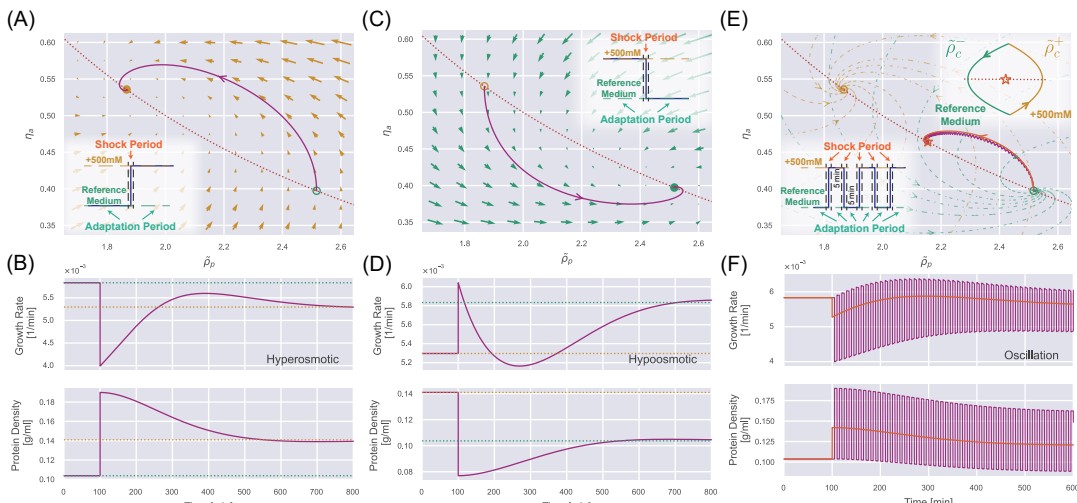

**Appendix 1—figure 2.** Dynamics of osmoresponse for unwalled cells. (**A**) The dynamics in the internal state space after a constant 500 mM hyperosmotic shock. The dotted curve represents $\tilde{\rho}_p\eta_a = 1$, and the solid trajectory is from numerical simulations. The arrows represent the stream flow described by *Equation 20* and *Equation 21*. The inset depicts the separation of timescale into shock periods and adaptation periods. (**B**) The temporal trajectory of growth rate (upper panel) and protein density (lower panel) corresponding to the 500 mM hyperosmotic shock in (**A**). The dashed lines represent the steady-state values in the reference medium (green) and the medium after perturbation (yellow). (**C**) The same analysis as (**A**) but for a constant 500 mM hypoosmotic shock. (**D**) The same analysis as (**B**) but for a constant 500 mM hypoosmotic shock. (**E**) The trajectory of the internal state during a 500 mM osmotic oscillation with a 10 min period (the purple line). The low inset depicts the separation of timescale into shock periods and adaptation periods. The upper inset shows that the trajectory finally converges to a periodic circle around a dynamic equilibrium point (marked by an orange star) on the red dotted curve $\tilde{\rho}_p\eta_a = 1$. The trajectory for a constant hyperosmotic shock with an amplitude of 222 mM (orange curve) almost coincides with the trajectory for the oscillator shock. (**F**) The purple curves represent the temporal trajectory of growth rate (upper panel) and protein density (lower panel) corresponding to the 500 mM osmotic oscillation in (**E**). The orange curves represent the corresponding ones for the constant hyperosmotic shock with an amplitude of 222 mM.

## 2. Oscillatory osmotic perturbation

In *Appendix 1—figure 2E*, we study the osmoregulation dynamics during an osmotic oscillation. The internal state trajectory approaches different targets during the hyperosmotic and hypoosmotic periods. Eventually, it reaches a periodic steady state around a dynamical equilibrium point on the curve $\tilde{\rho}_p\eta_a = 1$ (*Appendix 1—video 1*). In this case, the net changes of $\tilde{\rho}_p$ and $\eta_a$ are zero during one oscillation cycle:

$$\int_{\text{hyper}} \eta_r^+(1 - \tilde{\rho}_p\eta_a)\,\mathrm{d}t + \int_{\text{hypo}} \eta_r^-(1 - \tilde{\rho}_p\eta_a)\,\mathrm{d}t = 0, \tag{A15a}$$

$$\int_{\text{hyper}} \eta_r^+ \left[\left(\frac{\tilde{\rho}_p}{\tilde{\rho}_c^+}\right)^{H_a} - \eta_a\right]\,\mathrm{d}t + \int_{\text{hypo}} \eta_r^- \left[\left(\frac{\tilde{\rho}_p}{\tilde{\rho}_c^-}\right)^{H_a} - \eta_a\right]\,\mathrm{d}t = 0, \tag{A15b}$$

Here, the subscripts ± of $\eta_r$ stand for the hyperosmotic (500 mM) and hypoosmotic (the reference medium) period, respectively. As the external osmolarity switches, a sharp change occurs in $\eta_r = 1 - (\rho_p/\rho_c)^{H_r} = 1 - (\tilde{\rho}_p/\tilde{\rho}_c)^{H_r}$ due to the different $\tilde{\rho}_c$ values in the hyper/hypoosmotic periods.

We consider the limiting case where the oscillation period $T$ approaches 0. In this case, the values of $\tilde{\rho}_p$ and $\eta_a$ along the cycles can be well approximated by their values at the dynamical equilibrium point. Thus, we must have $\tilde{\rho}_p\eta_a = 1$ for the dynamical equilibrium point according to *Equation A15a*. This result suggests that the dynamic equilibrium point also lies on the curve for the steady states.

For unwalled cells, $\tilde{\rho}_c$ maintains a fixed value during the hyper/hypoosmotic period. After applying the same procedure to *Equation A15b*, the following relationship is derived,

$$(\eta_r^+ + \eta_r^-)\eta_a = \eta_r^+ \left(\frac{\tilde{\rho}_p}{\tilde{\rho}_c^+}\right)^{H_a} + \eta_r^- \left(\frac{\tilde{\rho}_p}{\tilde{\rho}_c^-}\right)^{H_a}. \tag{A16}$$

Therefore, the precise position of the dynamical equilibrium point $(\eta_a, \tilde{\rho}_p)$ can be determined from $\tilde{\rho}_p\eta_a = 1$ and **Equation A16**. In particular, the dynamic equilibrium point of a 500 mM oscillatory stimulus is equivalent to the equilibrium point of a 222 mM constant hyperosmotic shock. Surprisingly, the entire trajectories in the internal state space for an oscillatory stimulus and its equivalent hyperosmotic shock almost coincide (**Appendix 1—figure 2E**). Indeed, the protein density and growth rate followed by the equivalent constant hyperosmotic shock are essentially the time-averaged results of the oscillatory one (**Appendix 1—figure 2F**).

For walled cells under oscillatory perturbation, due to the extra dynamics of turgor pressure, the target equilibrium point $(\tilde{\rho}_c^{H_a/(H_a+1)}, \tilde{\rho}_c^{-H_a/(H_a+1)})$ moves along the curve $\tilde{\rho}_p\eta_a = 1$ (**Appendix 1—video 2** and **Appendix 1—figure 6**). The steady state in the internal state space is still a periodic circle around a dynamical equilibrium point on the curve $\tilde{\rho}_p\eta_a = 1$, similar to the case of unwalled cells.

## E. Cell-wall synthesis regulation is necessary for supergrowth

In this section, we discuss the conditions of the supergrowth phenomenon. Specifically, we focus on the scenario where the cell has undergone sufficient oscillatory osmotic cycles before switching back to the reference growth medium. Our conclusions are also valid for a constant hypoosmotic shock. By definition, the overall growth rate of the entire cytoplasmic volume is the weighted average of the dry-mass growth rate and the free-volume growth rate $\mu = (1 - f)\mu_r + f\mu_f$. We remark that the condition of supergrowth can be well approximated by the instantaneous condition: $\mu_f > \mu_r$ because $\mu_r$ only depends on the protein density, which changes much less significantly than the free-volume growth rate (**Appendix 1—figure 7**).

We rewrite the dynamics of the internal osmotic pressure, **Equation 18c**, in terms of the internal state $(\tilde{\rho}_p, \eta_a)$:

$$\dot{\Pi}_{in} = (\eta_a\tilde{\rho}_p\mu_r - \mu_f)\Pi_{in} \tag{A17}$$

We first discuss the case of an unwalled cell. After sufficient cycles of osmotic oscillation, the internal state $(\tilde{\rho}_p, \eta_a)$ converges to a periodic steady state that lies on $\tilde{\rho}_p\eta_a = 1$ (**Appendix 1—figure 2E**). After removing the oscillatory stimulus, the internal state leaves the curve $\tilde{\rho}_p\eta_a = 1$ and returns to this curve after fully adapting to the reference growth media. During this adaptation process, the internal osmotic pressure constantly equals the external one because the turgor pressure is zero. Therefore, setting $\dot{\Pi}_{in} = 0$ in **Equation A17**, the free-volume growth rate takes the form of

$$\mu_f = \mu_r\eta_a\tilde{\rho}_p. \tag{A18}$$

We note that the internal state $(\tilde{\rho}_p, \eta_a)$ always follows a counterclockwise trajectory (**Appendix 1—figure 2A and C**). Therefore, the unwalled cell inevitably enters the region of $\tilde{\rho}_p\eta_a < 1$ after removing the oscillatory stimulus (**Appendix 1—figure 3A**), and the supergrowth phenomenon cannot occur according to **Equation A18** (the upper panel of **Appendix 1—figure 3B**).

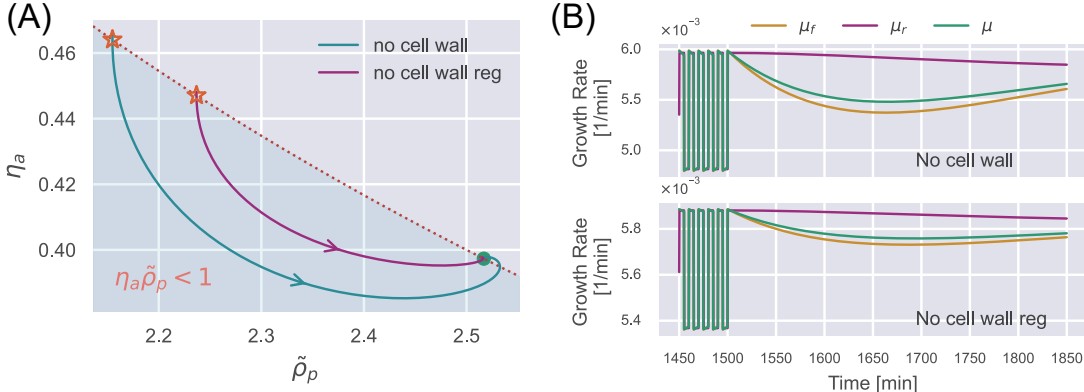

**Appendix 1—figure 3.** Dynamics of deficient cells after removing a sufficiently long oscillatory stimulus. (**A**) An oscillation stimulus is initially applied in the simulation until the cell reaches the periodic steady state (stars). The
*Appendix 1—figure 3 continued on next page*

*Appendix 1—figure 3 continued*

external osmolarity is then switched to the reference growth medium. The internal state evolves toward the steady state in the reference growth medium. Here, we show the results of an unwalled cell (blue curve) and a walled cell deficient in cell-wall synthesis regulation (purple curve). (**B**) The time dependence of the various growth rates corresponding to the simulations in (**A**).

Next, we discuss a walled cell deficient in cell-wall synthesis regulation. In this case, the growth rate of the relaxed cell-wall volume is always equal to the growth rate of dry mass, $\mu_{cw} = \mu_r$. According to *Equation 18d*, the time dependence of turgor pressure is given by

$$\dot{\sigma} = G\dot{\epsilon} = f(G + \sigma)(\mu_f - \mu_r). \tag{A19}$$

To maintain osmotic balance during the adaptation period, $\dot{\Pi}_{in} = \dot{\sigma}$, the free-volume growth rate must satisfy

$$\mu_f = \frac{f(G + \sigma) + \eta_a \tilde{\rho}_p \Pi_{in}}{f(G + \sigma) + \Pi_{in}} \mu_r. \tag{A20}$$

For the same reason as an unwalled cell, the internal state of a walled cell deficient in cell-wall synthesis regulation enters the region of $\tilde{\rho}_p \eta_a < 1$ after removing the oscillatory stimulus (*Appendix 1—figure 3A*). According to *Equation A20*, there is no supergrowth phenomenon after the oscillation stimulus (the lower panel of *Appendix 1—figure 3*). In summary, cell-wall synthesis regulation is necessary for supergrowth.

## F. The growth rate peak during the supergrowth phase

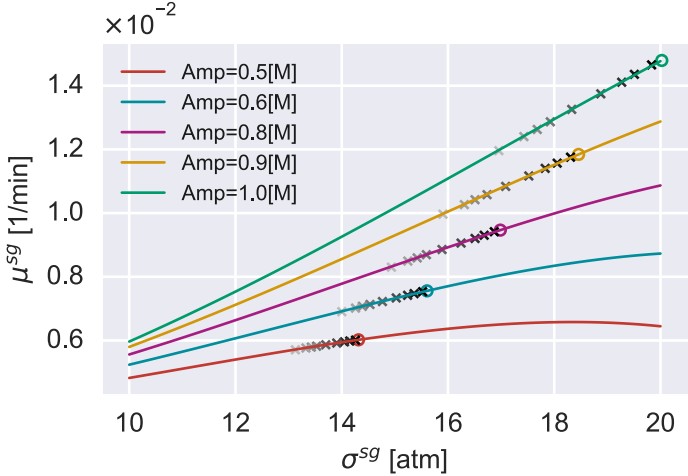

**Appendix 1—figure 4.** The relationship between the growth rate peak $\mu^{sg}$ and the turgor pressure at the growth rate peak $\sigma^{sg}$ under different amplitudes of hypoosmotic shock. The solid lines are the theoretical predictions, *Equation A23*. The open circles are obtained by approximating the turgor pressure as the immediate value after the shock $\sigma^f$ in *Equation A23*. The gray crosses represent the peak growth rates obtained by direct simulations using different $\tau_{cw}^+$. As the color goes from light to dark, $\tau_{cw}^+$ gradually decreases to zero.

Here, we consider the case of a constant hypoosmotic shock and derive an analytical expression of the growth rate peak $\mu^{sg}$. As discussed in the main text, the abrupt water influx right after the shock rapidly stretches the cell wall, resulting in elevated turgor pressure. One should note that the ratio of the dry-mass growth rate before and after the shock is equal to that of the relaxed cell-wall volume, both set by the crowding effect,

$$\frac{\mu_{cw}^f}{\mu_{cw}^i} = \frac{\mu_r^f}{\mu_r^i} = \frac{\eta_r^f}{\eta_r^i}. \tag{A21}$$

Meanwhile, the timescale $\tau_{cw}^+$ governing the upregulation of cell-wall synthesis efficiency $\eta_{cw}$ to its target value $(\sigma/\sigma_c)^{H_{cw}}$ is considerably smaller than the timescale associated with the osmoregulation process. Consequently, during the timescale of supergrowth, cells can be considered effectively deficient in osmoregulation, producing osmolyte molecules and dry mass at the same rate. Thus, we can simplify the time dependence of the internal osmotic pressure as $\dot{\Pi}_{in} = (\mu_r - \mu_f)\Pi_{in}$. Meanwhile, the water flux is physically constrained to ensure osmotic balance: $\dot{\Pi}_{in} = \dot{\sigma}$, leading to the expression:

$$\mu_f = \mu_r + \frac{1}{f + \dfrac{\Pi_{in}}{G+\sigma}}(\mu_{cw} - \mu_r). \tag{A22}$$

We point out that the timescales of the relaxation dynamics of $f$, $\Pi_{in}$, and $\sigma$ are all set by the doubling time (*Figure 3—figure supplement 1*); therefore, we can approximate them as constant for a short timescale. When the cell-wall synthesis efficiency $\eta_{cw} = \mu_{cw}/\mu_r$ reaches its target value, which is also its maximum value (*Figure 3*), the ratio $\mu_f/\mu_r$ reaches its maximum value according to *Equation A22*, giving rise to the growth rate peak $\mu^{sg}$. Substituting $\mu_{cw} = \mu_r(\sigma/\sigma_c)^{H_{cw}}$ into *Equation A22* and employing the definition $\mu = f\mu_f + (1-f)\mu_r$, we obtain the expression of $\mu_{sg}$:

$$\mu^{sg} = \mu_r\left\{1 + \frac{f}{f + \dfrac{\Pi_{in}}{\sigma+G}} \times \left[\left(\frac{\sigma}{\sigma_c}\right)^{H_{cw}} - 1\right]\right\}. \tag{A23}$$

All the variables on the right-hand side of *Equation A23* are calculated at the growth rate peak.

Owing to the gradual restoration of turgor pressure, the turgor pressure at the growth rate peak differs from the immediate value after the hypoosmotic shock. Through direct simulations, we acquire the precise value of $\sigma$ at the growth rate peak. Given the value of $\sigma$, the corresponding values of $\Pi_{in}$, $\mu_r$, and $f$ can be inferred. During the adaptation period, osmotic balance is maintained such that $\Pi_{in} = \Pi_{out}^f + \sigma$. Given $\Pi_{in}$, one can infer $\rho_p$ through *Equation A10* since osmoregulation is effectively inactive, from which we calculate the $\eta_r$ factor, and the dry-mass growth rate $\mu_r$ is subsequently derived from *Equation A6*. Consequently, $\sigma$ is the only degree of freedom on the right-hand side of *Equation A23*. We compare this analytical expression with direct simulations of *S. pombe* at different upregulation timescales of the cell-wall synthesis process (*Appendix 1—figure 4*). As $\tau_{cw}^+$ tends to zero, the growth rate peak can be well approximated by the immediate value after the hypoosmotic shock $\sigma^f$. This extreme case is a good approximation for WT cells, since the timescale of the turgor pressure recovery process is set by the doubling time (*Figure 3—figure supplement 1*), much longer than $\tau_{cw}^+$. Indeed, a strong correlation exists between the growth rate peak and the turgor pressure right after the shock (*Figure 3E*).

## G. The overshoot of turgor pressure after a single oscillation

During a single osmotic oscillation, the external osmolarity shifts to a hyperosmotic environment and then switches back to the reference growth medium (upper panel of *Appendix 1—figure 5*). We represent the turgor pressure jump resulting from the first osmolarity shift as $\sigma^{f,1} - \sigma^{i,1}$ and the corresponding jump from the second shift as $\sigma^{f,2} - \sigma^{i,2}$ (*Appendix 1—figure 5*). In *Figure 4E*, we show that the turgor pressure overshoot after the single oscillation $\sigma^{f,2} - \sigma_c$ can be approximated by the turgor recovery $\delta\sigma$ during the hyperosmotic period.

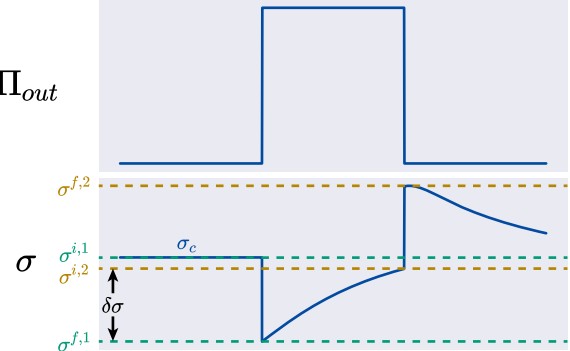

**Appendix 1—figure 5.** The upper panel shows the time dependence of the external osmolarity, and the lower panel shows the trajectory of the turgor pressure under a mild osmotic oscillation.

To prove the statement $\delta\sigma \approx \sigma^{f,2} - \sigma_c$, we only need to prove that the absolute magnitudes of the two turgor pressure jumps are nearly identical (**Appendix 1—figure 5**). The analytical expression for the turgor pressure after the jump is shown in **Equation A4**, where $\sigma^f$ is only related to $\sigma^i$ and $\epsilon^i$ before the shock. For a mild oscillation amplitude, the physiological properties of the cell are not much perturbed. In particular, the turgor pressure has almost recovered to the steady-state value $\sigma_c$ (**Appendix 1—figure 5**). We denote $\Delta\sigma = \sigma^f - \sigma^i$ and find that for a mild osmotic perturbation such that $\Delta\Pi_{in} = \Delta\sigma + \Delta\Pi_{out} \ll \Pi^i_{in} = \sigma^i + \Pi^i_{out}$, **Equation A4** can be simplified as

$$\frac{\Delta\sigma}{\sigma^i} = -\left(1 + G/\sigma^i\right)f^i\frac{\Delta\Pi_{out} + \Delta\sigma}{\Pi^i_{in}}. \tag{A24}$$

It is evident that $\Delta\sigma$ is directly proportional to $\Delta\Pi_{out}$. Given that $\Delta\Pi^2_{out} = -\Delta\Pi^1_{out}$, the turgor pressure jumps cancel each other out for a mild osmotic oscillation.

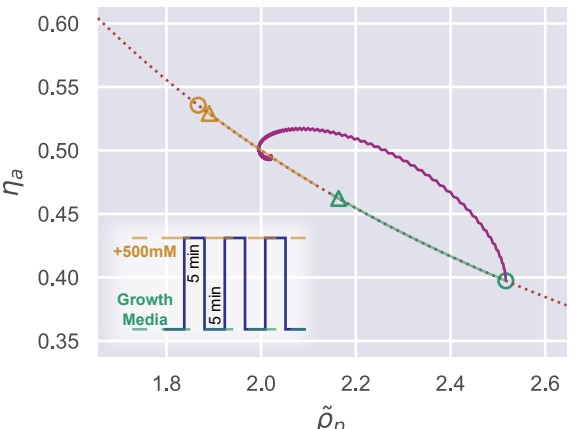

**Appendix 1—figure 6.** Trajectory of a wild-type walled cell in the internal state space during a 500 mM osmotic oscillation. During the oscillation, the equilibrium points move along the dotted curve $\tilde{\rho}_p\eta_a = 1$. The circles indicate the equilibrium points during the first oscillation cycle, while the triangles indicate their positions after a large enough number of oscillations.

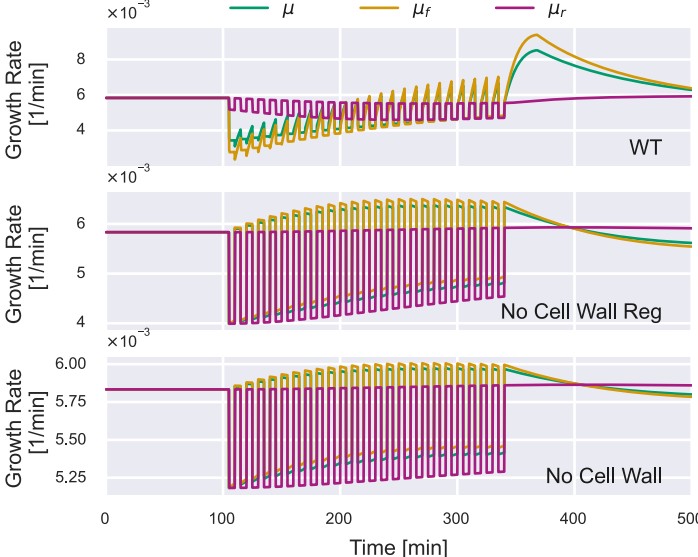

**Appendix 1—figure 7.** Time dependence of the free-volume growth rate $\mu_f$, the dry-mass growth rate $\mu_r$, and the overall growth rate $\mu$ under 500 mM osmotic oscillations. Here, we show the results of an intact wild-type (WT) cell (upper panel), a walled cell deficient in cell-wall synthesis regulation (middle panel), and an unwalled cell (lower panel).

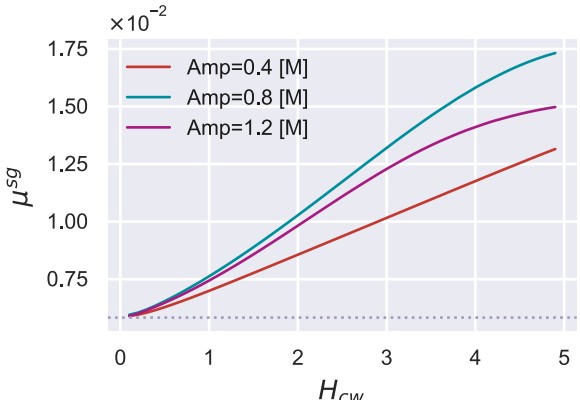

**Appendix 1—figure 8.** Given 24 cycles of osmotic oscillations with a period of 10 min, we vary the parameter $H_{cw}$, which dictates the sensitivity of cell-wall synthesis regulation. A higher growth rate peak $\mu^{sg}$ is observed as the sensitivity $H_{cw}$ increases, irrespective of the oscillation amplitude. The dotted line represents the growth rate in the reference growth medium.

