## [Editor Report · eLife Assessment]

This manuscript develops a theoretical model of osmotic pressure adaptation in microbes by osmolyte production and wall synthesis. The prediction of a rapid increase in growth rate on osmotic shock is experimentally validated using fission yeast. By using phenomenological rules rather than detailed molecular mechanisms, the model can potentially apply to a wide range of microbes, providing **important** insights that would be of interest to the wider community studying the regulation of cell size and mechanics. The level of coarse-graining and the assumptions and limitations of the model have been well described, providing a **convincing** foundation for making predictions. However, further experimental work on the validity of the core assumptions across a range of microbial organisms is needed to assess the universality of the model.

---

## [Referee Report · Reviewer #1 (Public review)]

Summary:

A theoretical model for microbial osmoresponse was proposed. The model assumes simple phenomenological rules: (i) the change of free water volume in the cell due to osmotic imbalance based on pressure balance, (ii) osmoregulation that assumes change of the proteome partitioning depending on the osmotic pressure that affects the osmolyte-producing protein production, (iii) the cell-wall synthesis regulation where the change of the turgor pressure to cell-wall synthesis efficiency to go back to the target turgor pressure, (iv) effect of intracellular crowding assuming that the biochemical reactions slow down for more crowding and stop when the protein density (protein mass divided by free water volume) reaches a critical value. The parameter values were found in the literature or obtained by fitting to the experimental data. The authors compare the model behavior with various microorganisms (*E. coli*, *B. subtils*, *S. cerevisiae*, *S. pombe*) and successfully reproduced the overall trend (steady state behavior for many of them, dynamics for *S. pombe*). In addition, the model predicts non-trivial behavior such as the fast cell growth just after the hypoosmotic shock, which is consistent with experimental observation. The authors further make experimentally testable predictions regarding mutant behavior and transient dynamics.

The theory assumes simple mechanistic dependence between core variables without going into specific molecular mechanisms of regulations. The simplicity allows the theory to apply to different organisms by adjusting the time scales with parameters, and the model successfully explains broad classes of observed behaviours. Mathematically, the model provides analytical expressions of the parameter dependencies and an understanding of the dynamics through the phase space without being buried in the detail. This theory can serve as a base to discuss the universality and diversity of microbial osmoresponse.

The coarse-grained nature of the model is the strength of the model in terms of its generality. However, it does not consider various regulations at the molecular level. Hence, certain adaptation features are not considered in the current version of the model. The updated manuscript discusses the pros and cons of the current approach.

---

## [Referee Report · Reviewer #2 (Public review)]

Summary:

In this study, Ye et al. have developed a theoretical model of osmotic pressure adaptation by osmolyte production and wall synthesis.

Strengths:

They validate their model predictions of a rapid increase in growth rate on osmotic shock experimentally using fission yeast. The study has several interesting insights which are of interest to the wider community of cell size and mechanics.

Comments on revisions:

The authors have in the revised manuscript addressed the aspects of the writing that were unclear. , that are listed previously as major and minor comments. We believe the issues raised by this reviewer have been adequately addressed in the manuscript.

---

## [Author Response]

The following is the authors’ response to the original reviews

List of major changes

(1) We have emphasized the assumptions underlying our modeling approach in the third paragraph of the Introduction section.

(2) We have included a new paragraph in the Discussion section to compare our model with a molecular mechanism-oriented model.

(3) We have included a new paragraph at the end of the Introduction section to outline the main content of each subsection in Results and the logical connections between them. Correspondingly, the chapter hierarchy and section titles have been adjusted.

(4) The Supplementary Material includes an additional table (Table S2) that provides detailed explanations of the symbols used in the model.

(5) We have included a new paragraph in the Introduction section to explicitly emphasize the phenomenological nature of our model and its broad applicability.

(6) In the Osmoregulation subsection, we have added a discussion on how our model can be directly generalized to scenarios involving the environmental uptake of osmolytes.

(7) We have included a more detailed examination of the limitations inherent in our modeling approach in the second last paragraph of the Discussion section.

(8) In the third last paragraph of the Discussion section, we have explicitly demonstrated that our model does not conflict with the observation that, in *E. coli*, cell wall synthesis is not directly regulated by the turgor pressure.

**Reviewer #1 (Public review):**
Summary:A theoretical model for microbial osmoresponse was proposed. The model assumes simple phenomenological rules: (i) the change of free water volume in the cell due to osmotic imbalance based on pressure balance, (ii) osmoregulation that assumes change of the proteome partitioning depending on the osmotic pressure that affects the osmolyte-producing protein production, (iii) the cell-wall synthesis regulation where the change of the turgor pressure to the cell-wall synthesis efficiency to go back to the target turgor pressure, (iv) effect of Intracellular crowding assuming that the biochemical reactions slow down for more crowding and stops when the protein density (protein mass divided by free water volume) reaches a critical value. The parameter values were found in the literature or obtained by fitting to the experimental data. The authors compare the model behavior with various microorganisms (*E. coli*, *B. subtils*, *S. cerevisiae*, *S. pombe*), and successfully reproduced the overall trend (steady state behavior for many of them, dynamics for *S. pombe*). In addition, the model predicts non-trivial behavior such as the fast cell growth just after the hypoosmotic shock, which is consistent with experimental observation. The authors further make experimentally testable predictions regarding mutant behavior and transient dynamics.Strength:The theory assumes simple mechanistic dependence between core variables without going into specific molecular mechanisms of regulations. The simplicity allows the theory to apply to different organisms by adjusting the time scales with parameters, and the model successfully explains broad classes of observed behaviors. Mathematically, the model provides analytical expressions of the parameter dependences and an understanding of the dynamics through the phase space without being buried in the detail. This theory can serve as a base to discuss the universality and diversity of microbial osmoresponse.

We would like to thank Reviewer 1 for thoroughly reading our work and appreciating our theoretical approach to investigating microbial osmotic response.

Weakness:The core part of this model is that everything is coupled with growth physiology, and, as far as I understand, the assumption (iv) (Eq. 8) that imposes the global reaction rate dependence on crowding plays a crucial role. I would think this is a strong and interesting assumption. However, the abstract or discussion does not discuss the importance of this assumption. In addition, the paper does not discuss gene regulation explicitly, and some comparison with a molecular mechanismoriented model may be beneficial to highlight the pros and cons of the current approach

We thank Reviewer 1 for their very helpful feedback. We have significantly revised the manuscript as suggested by Reviewer 1. See the detailed answers in the following.

**Reviewer #1 (Recommendations for the authors)**
(1) Explicitly stating the assumption (iv) in the abstract and discussing its role would help readers understand.

In the revised manuscript, we have significantly rewritten the third paragraph of the Introduction section to emphasize our key assumptions as suggested by Reviewer 1, including the relationship between global reaction rate and crowding:

“Our model assumes the following phenomenological rules: (1) the change in free water volume within the cell is driven by osmotic imbalance (Cadart et al., Nature Physics, 2019; Rollin et al., Elife, 2023), while the remaining volume changes in proportion to protein production; (2) osmoregulation influences the production of osmolyte-producing protein, governed by intracellular protein density (Scott et al., Science, 2010); (3) cell-wall synthesis is regulated through a feedback mechanism, wherein turgor pressure modulates the efficiency of cell-wall synthesis, enabling the cell to maintain a relatively stable turgor pressure; and (4) intracellular crowding slows down biochemical reactions as cytoplasmic density increases, with reactions ceasing entirely when protein density reaches a critical threshold.”

We have also modified the abstract to mention the crowding effects explicitly. Additionally, we have added a few sentences in the first and second paragraphs of the Discussion section to emphasize the importance of crowding effects to our conclusions regarding the growth rate reduction in steady states and the non-monotonic dependence of the growth rate peak on the shock amplitude after a hyperosmotic shock.

(2) I found [Shen W , Gao Z, Chen K, Zhao A, Ouyang Q, Luo C. The regulatory mechanism of the yeast osmoresponse under different glucose concentrations. *Iscience*. 2023 Jan 20;26(1)], which discusses the medium glucose concentration dependence of the response, focused on the gene regulatory circuit and the metabolic flux. As far as I understood, this paper considers the effect of the reallocation of resources but not the mechanical part of the osmoresponse such as pressure explicitly. It will be interesting to discuss the pros and cons in comparison with such a model. In principle, I will not be surprised if the current model does not differentiate the different glucose concentrations much since it is a more coarse-grained model, and I don't think it is a problem, but it will be good to have an explicit discussion.

We appreciate Reviewer 1's insightful comment regarding the work by Shen et al. (iScience, 2023), which elucidates the two distinct osmoresponse strategies in yeast. By quantifying Hog1 nuclear translocation dynamics and downstream protein expression, the study reveals that in a rich medium, cells can leverage surplus glycolytic products as defensive reserves, reallocating metabolic flux to facilitate rapid adaptation to osmotic changes. Conversely, limited glycolytic intermediates in low-glucose environments necessitate increased enzyme synthesis for osmotic adaptation.

The paper highlighted by Reviewer 1 studies yeast's adaptive strategies under two stresses— nutrient limitation and osmotic pressure and provides an important complement to our study.

In our simplified model, we did not include the interaction between cell growth and osmolyte production, assuming a constant fraction of ribosomes translating ribosomal proteins, supported by the experiments of *E. coli* (Dai et al., mBio, 2018). We remark that incorporating competitive dynamics for translational resources into our framework can be achieved by modifying the proportion of ribosomes translating themselves (*Xr*), from a constant to a function related to the translation strategy of the osmolyte-producing enzyme (*Xa*).

In the revised manuscript, we have included a new discussion in the third paragraph of the Discussion section to compare our approach with the molecular mechanism-oriented model:

“We remark that our model is intrinsically a coarse-grained model with many molecular details regarding gene expression regulation neglected, which allows us to gain more analytical insights. In [Shen et al., iScience, 2023], the authors studied the responses to osmotic stress in glucose-limited environments and found that cells exhibited stronger osmotic gene expression response under glucose-limited conditions than under glucose-rich conditions. Using a computational model based on molecular mechanisms combined with experimental measurements, the authors demonstrated that in a glucose-limited environment, glycolysis intermediates were limited, which required cells to express more glycerol-production enzymes for stress adaptation. In the current version of our model, we do not account for the interaction between cell growth and osmolyte production; instead, we assume a constant fraction of ribosomes dedicated to translating ribosomal proteins. Our model can be further generalized to include the more complex interactions, including the coupling between biomass and osmolyte production, e.g., by allowing the fraction of ribosomes translating (*Xr*) to depend on the translation strategy of the osmolyte-producing enzyme (*Xa*).”

(3) A minor comment: The authors call assumption (iii) (eq. 7) "positive feedback from turgor pressure to the cell-wall synthesis efficiency" (line 204). I have a hard time seeing this as positive feedback. It regulates the cell wall synthesis so that turgor pressure returns to the desired value; hence, isn't it negative feedback?

We apologize for this confusion. We have removed the term "positive feedback" in the revised manuscript.

**Reviewer #2 (Public review):**
Summary:In this study, Ye et al. have developed a theoretical model of osmotic pressure adaptation by osmolyte production and wall synthesis.Strengths:They validate their model predictions of a rapid increase in growth rate on osmotic shock experimentally using fission yeast. The study has several interesting insights which are of interest to the wider community of cell size and mechanics.Weaknesses:Multiple aspects of this manuscript require addressing, in terms of clarity and consistency with previous literature. The specifics are listed as major and minor comments.Major comments:(1) The motivation for the work is weak and needs more clarity.

We thank Reviewer 2 for this very helpful comment, which we believe has significantly improved our manuscript. We would like to clarify the two major motivations of our study.

First, we aim to construct a systems-level and coarse-grained model capable of elucidating the complex processes underlying microbial osmoresponse. By leveraging the separation of timescales associated with mechanical equilibrium, cell-wall synthesis regulation, and osmoregulation, our model facilitates in-depth analytical and numerical analysis of how these various processes interact during cellular adaptation. In particular, we demonstrate the key physiological functions of osmoregulation and cell-wall synthesis regulation.

Second, we seek to apply this model to interpret the phenomenon of supergrowth observed in fission yeast *Schizosaccharomyces pombe* (Knapp et al., Cell Systems, 2019). This application addresses an essential challenge in experimental studies: exclusive knockout experiments can be difficult, and mechanistic interpretations of experimental observations are often lacking. Our theoretical framework offers a valuable tool for understanding such phenomena, contributing to the fundamental knowledge of microbial physiology and developing predictive models for microbial behavior under osmotic stress.

In the revised manuscript, we have included a new paragraph at the end of the Discussion section to emphasize our motivations better:

“In this work, we construct a systems-level and coarse-grained model capable of elucidating the complex processes underlying microbial osmoresponse. By leveraging the separation of timescales associated with mechanical equilibrium, cell-wall synthesis regulation, and osmoregulation, our model facilitates in-depth analytical and numerical analysis of how these various processes interact during cellular adaptation. In particular, we demonstrate the key physiological functions of osmoregulation and cell-wall synthesis regulation. We then apply this model to interpret the unusual phenomenon of supergrowth observed in fission yeast. This application addresses an essential challenge in experimental studies: exclusive knockout experiments can be difficult, and mechanistic interpretations of experimental observations are often lacking. Our theoretical framework offers a valuable tool for understanding such phenomena, contributing to the fundamental knowledge of microbial physiology and developing predictive models for microbial behavior under osmotic stress.”

(2) The link between sections is very frequently missing. The authors directly address the problem that they are trying to solve without any motivation in the results section.

We are grateful to Reviewer 2 for their valuable feedback. In the revised manuscript, we have included a new paragraph at the end of the Introduction section to outline the main content of each subsection in Results and the logical connections between them:

“In the following “Results” section, we begin by outlining the primary assumptions and equations of our model in the subsection "Model Description," which includes four parts, each addressing one of the four phenomenological rules. Additional details can be found in Methods. We then proceed to the subsection “Steady states in constant environments”, where we employ our theoretical framework to analyze steady-state growth and examine how the growth rate varies with external osmolarity. In the “Transient dynamics after a constant osmotic shock” subsection, we investigate the time-dependent osmoresponse after a constant hyperosmotic and hypoosmotic shock. Finally, in “Comparison with experiments: supergrowth phenomena after osmotic oscillation”, we address the supergrowth phenomena observed in *S. pombe*, utilizing our model to elucidate these experimental observations.”

(3) The parameters used in the models (symbols) need to be explained better to make the paper more readable.

We apologize for this confusion. In the revised Supplementary Material, we have included an additional table (Table S2) to explain the meanings of the symbols employed in the model to help the reader better understand.

(4) Throughout the paper, the authors keep switching between organisms that they are modelling. There needs to be some consistency in this aspect where they mention what organism they are trying to model, since some assumptions that they make may not be valid for both yeast as well as bacteria.

We thank Reviewer 2 for this very helpful comment. We would like to clarify that our model is coarse-grained without including detailed molecular mechanisms; therefore, it presumably applies to various species of microorganisms. Indeed, the predicted steady-state growth curves derived from our model and the experimental data obtained from various organisms agree reasonably well (Figure 2A of the main text).

In the revised manuscript, we have explicitly emphasized the nature of our phenomenological model and its broad applicability in the fourth paragraph of the Introduction section:

“We remark that our model is coarse-grained, without including detailed molecular mechanisms, and is therefore applicable across diverse microbial species. Notably, the predicted steady-state growth rate as a function of internal osmotic pressure from our model aligns well with experimental data from diverse organisms. This alignment allows us to quantify the sensitivities of translation speed and regulation of osmolyte-producing protein in response to intracellular density. Additionally, we demonstrate that osmoregulation and cellwall synthesis regulation enable cells to adapt to a wide range of external osmolarities and prevent plasmolysis. Our model also predicts a non-monotonic time dependence of growth rate and protein density as they approach steady-state values following a constant osmotic shock, in concert with experimental observations (Rojas et al., PNAS, 2014; Rojas et al., Cell systems, 2017). Moreover, we show that a supergrowth phase can arise following a sudden decrease in external osmolarity, driven by cell-wall synthesis regulation, either through the direct application of a hypoosmotic shock or the withdrawal of an oscillatory stimulus. Remarkably, the predicted amplitudes of supergrowth (i.e., growth rate peaks) quantitatively agree with multiple independent experimental measurements.”

Furthermore, we have also included a comparison with a detailed molecular mechanism model in the third paragraph of the Discussion section:

“We remark that our model is intrinsically a coarse-grained model with many molecular details regarding gene expression regulation neglected, which allows us to gain more analytical insights. In [Shen et al., iScience, 2023], the authors studied the responses to osmotic stress in glucose-limited environments and found that cells exhibited stronger osmotic gene expression response under glucose-limited conditions than under glucose-rich conditions. Using a computational model based on molecular mechanisms combined with experimental measurements, the authors demonstrated that in a glucose-limited environment, glycolysis intermediates were limited, which required cells to express more glycerol-production enzymes for stress adaptation. In the current version of our model, we do not account for the interaction between cell growth and osmolyte production; instead, we assume a constant fraction of ribosomes dedicated to translating ribosomal proteins. Our model can be further generalized to include the more complex interactions, including the coupling between biomass and osmolyte production, e.g., by allowing the fraction of ribosomes translating (*Xr</supb*) to depend on the translation strategy of the osmolyte-producing enzyme (*Xa*).”

(5) The extent of universality of osmoregulation i.e the limitations are not very well highlighted.

The osmoregulation mechanism described in our model primarily addresses changes in cytoplasmic osmolarity through the de-novo synthesis of compatible solutes, widely observed across bacteria, archaea, and eukaryotic microorganisms. This review article (GundeCimerman et al., FEMS microbiology reviews, 2018) provides an extensive summary and exploration of the primary compatible solutes utilized by organisms from all three domains of life, underscoring the prevalence of this osmoregulatory strategy. Furthermore, our model can be directly generalized to scenarios involving the direct uptake of osmolytes from the environment. One only needs to change the interpretation of the parameter, *𝑘𝑎* in the production of osmolyte molecule, \begin{document}$\dot{N}_{\alpha}=k_{\alpha} m_{p, a}$\end{document}, from the synthesis rate to the uptake rate, and all the results are equally applicable. In the revised manuscript, we have briefly discussed this point in the subsection “Osmoregulation.”

We agree with Reviewer 2 that our model's coarse-grained nature makes it broadly applicable to diverse microbial taxa; however, more specialized adaptations are beyond our model. In the revised manuscript, we have included a more detailed examination of the limitations inherent in our modeling approach in the second last paragraph of the Discussion section:

“We remark several limitations of our current coarse-grained model. First, the high membrane tension that inhibits transmembrane flux of peptidoglycan precursors, leading to a growth inhibition before the supergrowth peak (Rojas et al., Cell systems 2017) is beyond our model. Second, in our current framework, the osmoregulation and cell-wall synthesis regulation rely on the instantaneous cellular states. However, microorganisms can exhibit memory effects to external stimuli by adapting to their temporal order of appearance (Mitchell et al., Nature 2009). Notably, in the osmoregulation of yeast, a short-term memory, facilitated by post-translational regulation of the trehalose metabolism pathway, and a long-term memory, orchestrated by transcription factors and mRNP granules, have been identified (Jiang et al., Science signaling 2020). Besides, our model does not account for the role of osmolyte export in osmoregulation (Tamas et al., Molecular microbiology, 1999) and the interaction between biomass and osmolyte production (Shen et al., Iscience 2023). Extending our model to include more realistic biological processes will be interesting.”

(6) Line 198-200: It is not clear in the text what organisms the authors are writing about here. "Experiments suggested that the turgor pressure induce cell-wall synthesis, e.g., through mechanosensors on cell membrane [45, 46], by increasing the pore size of the peptidoglycan network [5], and by accelerating the moving velocity of the cell-wall synthesis machinery [31]". This however is untrue for bacteria as shown by the study reference 22 is this paper: E. Rojas, J. A. Theriot, and K. C. Huang, Response of *Escherichia coli* growth rate to osmotic shock, Proceedings of the National Academy of Sciences 111, 7807 (2014).

We thank Reviewer 2 for pointing out this very important issue and apologize for the confusion. References 45 and 46 (Dupres et al., Nature Chemical Biology 2009; Neeli-Venkata et al., Developmental Cell 2021) discuss how Wsc1 acts as a mechanosensor in *S. pombe*, detecting turgor pressure and activating pathways that reinforce the cell wall. Reference 5 (Typas et al., Cell 2010) explains the role of LpoA and LpoB, the two outer membrane lipoprotein regulators in *E. coli*, which modulate peptidoglycan synthesis in an extracellular manner. Reference 31 (Amir and Nelson, PNAS 2012) is a theoretical paper showing that turgor pressure may accelerate the moving velocity of the cell wall synthesis machinery in *E. coli*. In the revised manuscript, we have been more explicit about the organisms we refer to in the subsection “Cell-wall synthesis regulation.”

Meanwhile, we agree with Reviewer 2 that cell wall synthesis may not be directly regulated by turgor pressure in *E. coli* (Rojas et al., PNAS 2014). We would like to clarify that this scenario is also included in our model corresponding to *Hcw* = 0 (Eq. (7) in the main text): the turgor pressure does not affect the cell-wall synthesis. Therefore, the supergrowth phenomenon observed in *S. pombe* does not manifest under hypotonic stimulation in *E. coli.*

In the revised manuscript, we have emphasized this point more explicitly in the third last paragraph of the Discussion section:

“Reference 22 (Rojas et al., PNAS, 2014) showed that the expansion of *E. coli* cell wall is not directly regulated by turgor pressure, and this scenario is also included in our model as the case of *Hcw* = 0. According to our model, the supergrowth phase is absent if *Hcw* = 0 (Figure S8), consistent with the absence of a growth rate peak after a hypoosmotic shock in the experiments of *E. coli* (Rojas et al., PNAS, 2014). Meanwhile, our predictions are consistent with the growth rate peak after a hypoosmotic shock observed for *B. subtilis* (Rojas et al., Cell systems, 2017).”

(7) The time scale of reactions to hyperosmotic shocks does not agree with previous literature (reference 22). Therefore defining which organism you are looking at is important. Hence the statement " Because the timescale of the osmoresponse process, which is around hours (Figure 3B), is much longer than the timescale of the supergrowth phase, which is about 20 minutes, the turgor pressure at the growth rate peak can be well approximated by its immediate value after the shock." from line 447 does not seem to make sense. The authors need to address this.

We apologize for this confusion. In the revised manuscript, we have clarified that the cited time scales are for the fission yeast *S. pombe* after Eq. (13) in the main text.

**Reviewer #2 (Recommendations for the authors):**
(1) Inconsistency in nomenclature: On line 117, the equation reads *Vb* = *αmp* where *Vb* is the bound volume. Whereas bound volume has been referred to as *Vbd* previously and in Figure 1.

Answer: We apologize for this confusion. In our model, the total bound volume*Vb* comprises the volume of dry mass and bound water, *Vb* = *Vbd* + *Vbw*, where *Vbd* is the volume occupied by dry mass and *Vbw* is the volume of bound water. In the revised manuscript, we have added a brief discussion of this point in the caption of Figure 1.

(2) Line 180: Please define 𝜌𝜌 for equation 4.

We apologize for this confusion. In the text, the symbol *𝜌p* denotes the mass of a given substance per unit volume of free water, and its unit is g/ml. The specific substance in consideration is indicated by a subscript. For example, *𝜌p* in Eq. (4) represents the protein density, and *𝜌c* stands for the critical protein density, above which intracellular chemical reactions cease according to Eq. (8) of the main text. In the revised manuscript, we have clarified the meaning of *𝜌c* after Eq. (4).

(3) Line 187: Equation 5 also needs to be explained better. Hence there is a need to be more specific while stating the assumptions.

The elastic modulus 𝐺 defined in Eq. (5) of the main text is a measure of the cell wall's resistance to volume expansion. We assume a constant 𝐺 for simplicity, which is reasonable when the cell wall deformation is mild. In the revised manuscript, we have been more explicit about our assumptions regarding the turgor pressure in the subsection “Cell-wall synthesis regulation.”

(4) Line 225: For a biological audience some elaboration on "glass transition" may be required- either as a reference to a review or to a 1 sentence statement of relevance.

We appreciate Reviewer 2’s helpful comment. In the revised manuscript, we have added a brief introduction to the glass transition and a citation to a review paper (Hunter and Weeks, Rep. Prog. Phys. 2012) at the beginning of the subsection “Intracellular crowding.”

(5) Line 247: "All growth rates in steady states of cell growth are the same: *𝜇𝑓* = *𝜇r* = *𝜇cw*". The authors need to explain in a line or two why this is true. Since the processes are independent, it is safe to assume that all 𝜇's are constant, but it is not obvious why they should all be equal.

We apologize for the lack of a clear explanation regarding the equality of steady-state growth rates in our previous manuscript. In the revised manuscript, we have added a brief explanation of the equality of the three growth rates at the beginning of the subsection “Steady states in constant environments”:

“When cell growth reaches a steady state, the proportions of all components, including free water volume, cell mass, and cell wall volume, must be constant relative to the total cell volume to ensure homeostasis. Therefore, all growth rates in steady states of cell growth must be the same: *𝜇𝑓* = *𝜇r* = *𝜇cw*.”

(6) Line 264: "Because the typical doubling times of microorganisms are around hours, we can estimate *𝜇𝑓*/*kw* ∼ 10 Pa [51, 52] ..." since the authors are generalizing for yeast and bacteria, specifically *E. coli*, this is not a valid assumption to make. There is also a need to explain the basis of "*𝜇𝑓*/*kw* ∼ 10 Pa".

We appreciate the need for clarity in the estimation and its implications. The rough estimation of *𝜇𝑓*/*kw* ~ 10 Pa in the main text is given by:\begin{document}$$\displaystyle \frac{\mu_{f}}{k_{w}}=\frac{100^{-1}[\mathrm{min}]^{-1}}{100[\mathrm{min}]^{-1}[\mathrm{atm}]^{-1}}=10^{-4}[{atm}]=10[\mathrm{Pa}]$$\end{document}

Here, the typical value of *𝜇𝑓* (which equals to *𝜇r* in steady state) is approximated by the inverse of the cell cycle, which is around hours. The estimation above is employed to justify the assumption that *𝜇𝑓*/*kw* is much smaller than the cytoplasmic osmotic and turgor pressures, which can be several atmospheric pressures.

For the case of *E. coli*, based on the experimental results from Boer et al. (Boer et al., Biochemistry 2011), an 800mM hypoosmotic shock leads to a rapid expansion of cell volume accomplished within a time scale of 0.1s, from which we obtain:\begin{document}$$\displaystyle \frac{\mu_{f}}{k_{w}}=\frac{20^{-1}[\mathrm{min}]^{-1}}{\frac{0.1^{-1}[\mathrm{s}]^{-1}}{0.8 * 24.93[\mathrm{atm}]}}=1.66 * 10^{-3}[\mathrm{atm}]$$\end{document}

Therefore, our assumption that *𝜇𝑓*/*kw* is much smaller than the cytoplasmic osmotic and turgor pressures is still valid.

In the revised manuscript, we have increased the estimation ranges to include the case of *E. coli* in the first paragraph of the subsection “Steady states in constant environments.”

(7) Lines 279-283 need to be explained better.

We apologize for the confusion. In the revised manuscript, we have explained more explicitly the meaning of the growth curve in the second paragraph of the subsection “Steady states in constant environments”:

“Intriguingly, the relationship between the normalized growth rate (\begin{document}$\mu_{r} / \mu_{r}^{m a x}$\end{document}) and the normalized cytoplasmic osmotic pressure (\begin{document}$\Pi_{i n} / \Pi_{i n, c}$\end{document}), which we refer to as the growth curve in the following, has only one parameter *𝐻r*/(*𝐻𝑎*) . Therefore, the growth curves of different organisms can be unified by a single formula, Eq. (10b), and different organisms may have different values of *𝐻r*/(*𝐻𝑎* + 1).”

(8) In Figure 3, an arrow representing the onset of osmotic shock would make the figure more intuitive to understand.

We appreciate Reviewer 2 for this helpful suggestion. We have modified Figure 3 as suggested.

(9) It is unclear to me if the growth rate 𝜇𝜇𝑟𝑟 is representative of the growth of total protein. This can be motivated better.

We would like to clarify that the growth rate 𝜇𝜇𝑟𝑟 is defined as the changing rate of total protein mass divided by the total protein mass:\begin{document}$$\displaystyle \mu_{r} \equiv \frac{\dot{m}_{p}}{m_{p}}=\frac{k_{r} m_{p, r}}{m_{p}}=k_{r} \phi_{r} .$$\end{document}

Here, *𝑚𝑝,𝑟* is the total mass of ribosomal proteins and 𝑘𝑘𝑟𝑟 is a constant proportional to the elongation speed of ribosome. The expression of *𝜇𝑟* is a direct consequence of ribosomes being responsible for producing all proteins. In the revised manuscript, we have added more details in the introduction of the variable *𝜇𝑟* in the last paragraph of the subsection “Cell growth”:

“In this work, we assume that the dry-mass growth rate is proportional to the fraction of ribosomal proteins within the total proteome for simplicity, *𝜇𝑟* = *𝑘r𝑚𝑝,𝑟*/*𝑚𝑝* = *𝑘r𝜙𝑟*. This assumption leverages the fact that ribosomes are responsible for producing all proteins. The proportionality coefficient *𝑘𝑟* encapsulates the efficiency of ribosomal activity, being proportional to the elongation speed of the ribosome. We remark that 𝑘𝑘𝑟𝑟 is influenced by the crowding effect, which we address later.”